

# Size-resolved Composition and Morphology of Particulate Matter During the Southwest Monsoon in Metro Manila, Philippines

Melliza Templonuevo Cruz[1,2], Paola Angela Bañaga[1,3], Grace Betito[3], Rachel A. Braun[4], Connor Stahl[4], Mojtaba Azadi Aghdam[4], Maria Obiminda Cambaliza[1,3], Hossein Dadashazar[4], Miguel Ricardo Hilario[3], Genevieve Rose Lorenzo[1], Lin Ma[4], Alexander B. MacDonald[4], Preciosa Corazon Pabroa[5], John Robin Yee[5], James Bernard Simpas[1,3], Armin Sorooshian[4,6]

[1]Manila Observatory, Quezon City 1101, Philippines
[2]Institute of Environmental Science and Meteorology, University of the Philippines, Diliman, Quezon City 1101, Philippines
[3]Department of Physics, School of Science and Engineering, Ateneo de Manila University, Quezon City 1101, Philippines
[4]Department of Chemical and Environmental Engineering, University of Arizona, Tucson, AZ, USA
[5]Philippine Nuclear Research Institute, Commonwealth Avenue, Diliman, Quezon City 1101, Philippines
[6]Department of Hydrology and Atmospheric Sciences, University of Arizona, Tucson, AZ, USA

*Correspondence to:* Melliza Templonuevo Cruz (liz@observatory.ph)



**Abstract**
This paper presents novel results from size-resolved particulate matter (PM) mass, composition,
and morphology measurements conducted during the 2018 Southwest Monsoon (SWM) season in
Metro Manila, Philippines. Micro-Orifice Uniform Deposit Impactors (MOUDIs) were used to
collect PM sample sets that were analyzed for mass, morphology, black carbon (BC), and
composition of the water-soluble fraction. The bulk of the PM mass was between 0.18–1.0 μm
with a dominant mode between 0.32–0.56 μm. Similarly, most of the black carbon (BC) mass was
found between 0.10–1.0 μm (the so-called Greenfield gap), peaking between 0.18–0.32 μm, where
wet scavenging by rain is inefficient. In the range of 0.10 – 0.18 μm, BC constituted 78.1% of the
measured mass. Comparable contributions of BC (26.9%) and the water-soluble fraction (31.3%)
to total PM were observed and most of the unresolved mass, which in total amounted to 41.8%,
was for diameters exceeding 0.32 μm. The water-soluble ions and elements exhibited an average
combined concentration of 8.53 μg m$^{-3}$, with $SO_4^{2-}$, $NH_4^+$, $NO_3^-$, $Na^+$, and $Cl^-$ as the major
contributors. Positive Matrix Factorization (PMF) was applied to identify the possible aerosol
sources and estimate their contribution to the water-soluble fraction of collected PM. The factor
with the highest contribution was attributed to "Aged/Transported" aerosol (48.0%) while "Sea
Salt" (22.5%) and "Combustion" emissions (18.7%) had comparable contributions.
"Vehicular/Resuspended Dust" (5.6%) as well as "Waste Processing" emissions (5.1%) were also
identified. Microscopy analysis highlighted the ubiquity of non-spherical particles regardless of
size, which is significant when considering calculations of parameters such as single scattering
albedo, asymmetry parameter, and extinction efficiency.
Results of this work have implications for aerosol impacts on public health, visibility, and regional
climate as each of these depend on physicochemical properties of particles as a function of size.
The significant influence from Aged/Transported aerosol to Metro Manila during the SWM season
indicates that local sources in this megacity do not fully govern this coastal area's aerosol
properties and that PM in Southeast Asia can travel long distances regardless of the significant
precipitation and potential wet scavenging that could occur. That the majority of the regional
aerosol mass burden is accounted for by BC and other insoluble components has important
downstream effects on the aerosol hygroscopic properties, which depend on composition. The
results are relevant for understanding the impacts of monsoonal features on size-resolved aerosol
properties, notably aqueous processing and wet scavenging. Finally, the results of this work



provide contextual data for future sampling campaigns in Southeast Asia such as the airborne
component of the Cloud, Aerosol, and Monsoon Processes Philippines Experiment (CAMP$^2$Ex)
planned for the SWM season in 2019. Aerosol characterization via remote-sensing is notoriously
difficult in Southeast Asia, which elevates the importance of datasets such as the one presented
here.



## 1. Introduction

Ambient atmospheric aerosol particles impact human health, visibility, climate, and the hydrological cycle. Major factors governing these behaviors, such as deposition fraction in the respiratory system and activation into cloud condensation nuclei (CCN), include size and chemical composition. Therefore, size-resolved measurements of ambient aerosol particles can lend additional insights to the behaviors and implications of particulate matter (PM) in the atmosphere. One region of interest for characterization of aerosols is Southeast Asia due to increasing urbanization and the exposure of the population to a variety of aerosol sources, both natural and anthropogenic (Hopke et al., 2008). However, use of space-borne remote-sensing instrumentation presents a challenge for characterization of aerosol in this region, due to issues such as varying terrain and cloud cover (Reid et al., 2013).

The Philippines represents a country in Southeast Asia with a developing economy, rapid urbanization, old vehicular technology, and less stringent air quality regulations (e.g., Alas et al., 2017). It is also highly sensitive to the effects of climate change including prolonged dry periods and reductions in southwest monsoon (SWM) rainfall in recent decades (e.g., Cruz et al., 2013). Metro Manila is the country's capital and center of political and economic activities. Also referred to as the National Capital Region, Metro Manila is composed of 16 cities and a municipality that collectively occupy a land area of ~619 km$^2$. As of 2015, Metro Manila had a population of approximately 12.88 million (Philippine Statistics Authority, 2015). Of the cities comprising the Metro Manila area, the one that is the focus of this study, Quezon City, is the most populated (2.94 million people) with a population density of ~17,000 km$^{-2}$ as of 2015 (Philippine Statistics Authority, 2015).

The rainfall pattern in Southeast Asia is governed by topographic effects and the prevailing surface winds brought by the monsoons. Mountain ranges in the Philippines are generally oriented north to south in the eastern and western coasts. As such, northeasterly winds during the East Asian winter monsoon that starts in November brings wetness (dryness) on the eastern (western) coasts of the country. In contrast, the rainy season starts in May when the Western North Pacific subtropical high moves northeast and the Asian summer monsoon enables the propagation of southwesterly wind through the Philippines (Villafuerte et al., 2014). Metro Manila, located on the western side of the Philippines, therefore experiences wet (May-October) and dry (November-April) seasons. The large seasonal shift in prevailing wind directions can cause changes in the





source locations of aerosol transported to the Philippines and the subsequent direction in which
emissions from the Philippines are transported, such as to the northwest (e.g., Chuang et al., 2013)
or southwest (e.g., Farren et al., 2019). However, one interesting feature of Metro Manila is the
consistency of $PM_{2.5}/PM_{10}$ mass concentrations during both the dry (44/54 µg m$^{-3}$) and wet seasons
(43/55 µg m$^{-3}$) (Kim Oanh et al., 2006), which stands in contrast to typical assumptions that
increased wet scavenging during rainy seasons would lead to decreases in measured PM (e.g., Liao
et al., 2006).  While similar results are observed in Chennai, India, this behavior is different than
other cities in Asia, including Bandung City (Indonesia), Bangkok (Thailand), Beijing (China),
and Hanoi City (Vietnam), that exhibit reduced $PM_{2.5}$ levels during the wet season as compared to
the dry season (Kim Oanh et al., 2006). While the total PM levels may stay constant across the wet
and dry seasons, seasonally-resolved analyses will provide additional insights into how the
composition, morphology, and sources (transported vs. local emissions) change on a seasonal
basis.

Metro Manila has been drawing growing interest for PM research owing to the significant

levels of black carbon (BC). A large fraction of PM in Metro Manila can be attributed to BC (e.g.,
~50% of $PM_{2.5}$; Kim Oanh et al., 2006), with previously measured average values of BC at MO
reaching ~10 µg m$^{-3}$ for $PM_{2.5}$ (Simpas et al., 2014). The impacts of the high levels of BC present
on human health have also received attention (Kecorius et al., 2019). Identified major sources of
BC include vehicular, industrial, and cooking emissions (Bautista et al., 2014; Kecorius et al.,
2017). Vehicular emissions, especially along roadways where personal cars and motorcycles,
commercial trucks, and motorized public transportation, including powered tricycles and *jeepneys*,
are plentiful. For instance, measurements of $PM_{2.5}$ at the National Printing Office (NPO) located
alongside the major thoroughfare Epifanio de los Santos Avenue (EDSA) were on average 72 µg
m$^{-3}$; this value is twice the average concentration at the Manila Observatory (MO), an urban mixed
site located approximately 5 km from NPO (Simpas et al., 2014).  In addition to local emissions,
long-range transport of pollution, such as biomass burning, can also impact the study region (e.g.,
Xian et al., 2013; Reid et al., 2016a/b). However, most past work referenced above has focused on
either total $PM_{2.5}$ or $PM_{10}$ composition, and therefore, detailed size-resolved composition
information has been lacking in this region. Like other monsoonal regions (Crosbie et al., 2015;
Qu et al., 2015), it is of interest for instance to know if products of aqueous processing (e.g., sulfate,
organic acids) during the monsoonal period, promoted by the high humidity, become more





prominent in certain size ranges to ultimately enhance hygroscopicity, which is otherwise
suppressed with higher BC influence.
A year-long sampling campaign (Cloud, Aerosol, and Monsoon Processes Philippines
Experiment (CAMP$^2$Ex) weatHEr and CompoSition Monitoring (CHECSM) study) was
established in July 2018 to collect size-resolved aerosol measurements in Metro Manila. The aim
of this study is to report size-resolved PM measurements taken over the course of the SWM (July-
October) of 2018 in Quezon City, Metro Manila, Philippines as part of CHECSM. The results of
this study are important for the following reasons: (i) they provide size-resolved analysis of BC in
an area previously characterized as having one of the highest BC mass percentages in the whole
world; (ii) they provide a basis for better understanding the unusual phenomenon of having similar
PM levels during a wet and dry season; (iii) they provide contextual data for contrasting with both
other coastal megacities and also other monsoonal regions; and (iv) they can lend insights into the
characteristics of aerosol transported both into and out of Metro Manila and how important local
sources are in Metro Manila relative to transported pollution. Outcomes of this study include (i)
the first size-resolved characterization of both aerosol composition and morphology in Metro
Manila for the SWM, with implications in terms of PM effects on climate, visibility, the
hydrological cycle, and public health owing to the dependence of these impacts on particle size;
(ii)  archival data that contributes to the timeline of aerosol research in Metro Manila, and more
broadly Southeast Asia, where there is considerable concern over air pollution; and (iii) baseline
data for aerosol composition to be used to inform and assist research to be conducted during future
field campaigns in Southeast Asia including the same seasonal period (i.e., SWM) in 2019 as part
of CAMP$^2$Ex, which will involve both surface and airborne measurements.
**2. Experimental Methods**
**2.1 Sample Site**
Sampling was performed at MO in Quezon City, Philippines (14.64° N, 121.08° E). The
sampling instrumentation was located on the 3rd floor of the MO office building (~85 m above
sea level). Figure 1 visually shows the sampling location and potential surrounding sources. Past
work focused on PM$_{2.5}$ suggested that the study location is impacted locally mostly by traffic,
various forms of industrial activity, meat cooking from local eateries, and, based on the season,
biomass burning (Cohen et al., 2009). Fourteen sample sets were collected during the SWM season





(July-October 2018), with details about the operational and meteorological conditions during each
sample set shown in Table 1. Meteorological data were collected using a Davis Vantage Pro 2 Plus
weather station co-located with the aerosol measurements at MO. Except for precipitation, which
is reported here as accumulated rainfall, reported values for each meteorological parameter
represent averages for the sampling duration of each aerosol measurement.
The mean temperature during the periods of MOUDI sample collection ranged from 24.9
to 28.1° C, with accumulated rainfall ranging widely from no rain to up to 78.4 mm. To identify
sources impacting PM via long-range transport to the Metro Manila region, Figure 1a summarizes
the five-day back-trajectories for air masses arriving at MO on the days when samples were being
collected, calculated using the NOAA Hybrid Single-Particle Lagrangian Integrated Trajectory
(HYSPLIT) model (Stein et al., 2015; Rolph, 2016).  Trajectory calculations were started at 00,
06, 12, and 18 hours in MO at the height of the MOUDI inlet using meteorological files from the
NCEP/NCAR Reanalysis dataset. Trajectory cluster analysis was conducted using TrajStat (Wang
et al., 2009). The back-trajectories in Figure 1a show that indeed 66% of the wind came from the
southwest during the sampling periods.
**2.2 MOUDI Sample Sets**
Particulate matter was collected on Teflon substrates (PTFE membrane, 2 μm pore, 46.2
mm, Whatman) in Micro-Orifice Uniform Deposit Impactors (MOUDI, MSP Corporation, Marple
et al., 2014). Size-resolved measurements were taken at the following aerodynamic cutpoint
diameters ($D_p$): 18, 10, 5.6, 3.2, 1.8, 1.0, 0.56, 0.32, 0.18, 0.10, 0.056 μm. For a subset of the
sampling periods, two pairs of MOUDI sets were collected simultaneously such that both sets in
each pair could undergo different types of analyses. One set in each pair underwent gravimetric
analysis using a Sartorius ME5-F microbalance. MOUDI set 13 was additionally examined with a
Multi-wavelength Absorption Black Carbon Instrument (MABI; Australian Nuclear Science and
Technology Organisation). This optically-based instrument quantifies absorption and mass
concentrations at seven wavelengths between 405 and 1050 nm; however, results are reported only
for 870 nm to be consistent with other studies as BC is the predominant absorber at that wavelength
(e.g., Ramachandran and Rajesh, 2007; Ran et al., 2016). One additional sample set for microscopy
analysis was collected for one hour on August 1 using aluminum substrates.
**2.3 Chemical Composition Analysis**



In order to preserve samples for additional analysis, each Teflon substrate was cut in half.
A half of each substrate was extracted in 8 mL of Milli-Q water (18.2 MΩ-cm) through sonication
for 30 min in a sealed polypropylene vial. A blank substrate was processed in the same method to
serve as a background control sample. Subsequent chemical analysis of the water-soluble
components in the aqueous extracts were performed using ion chromatography (IC; Thermo
Scientific Dionex ICS - 2100 system) for the following species: cations = $Na^+$, $NH_4^+$, $Mg^{2+}$, $Ca^{2+}$,
dimethylamine (DMA), trimethylamine (TMA), diethylamine (DEA); anions =, methanesulfonate
(MSA), pyruvate, adipate, succinate, maleate, oxalate, phthalate, $Cl^-$, $NO_3^-$, $SO_4^{2-}$. Owing to co-
elution of TMA and DEA in the IC system, a cumulative sum of the two is reported here, which
represents an underestimate of their total mass concentration owing to overlap in parts of their
peaks. Limits of detection (LOD) were calculated for each species based on their respective
calibration curve (Table S1), with LOD being three times the standard deviation of the residuals
(predicted signal minus measured signal) divided by the slope of the calibration curve (Miller and
Miller, 2018).
The aqueous extracts were simultaneously characterized for elemental composition using
triple quadrupole inductively coupled plasma mass spectrometry (ICP-QQQ; Agilent 8800 Series)
for the following species: K, Al, Fe, Mn, Ti, Ba, Zn, Cu, V, Ni, P, Cr, Co, As, Se, Rb, Sr, Y, Zr,
Nb, Mo, Ag, Cd, Sn, Cs, Hf, Tl, Pb. Limits of detection of the examined elements were calculated
automatically by the ICP-QQQ instrument and were in the ppt range (Table S1). The sample
concentrations represent an average of three separate measurements with a standard deviation of
3% or less.
Note that some species were detected by both IC and ICP-QQQ (i.e., $Na^+$, $K^+$, $Mg^{2+}$, $Ca^{2+}$),
and that the IC concentrations are used here for all repeated species with the exception of $K^+$ owing
to better data quality from ICP-QQQ. All IC and ICP-QQQ species concentrations for samples
have been corrected by subtracting concentrations from background control samples.
**2.4 Microscopy Analysis**
As already noted, one MOUDI set on August 1 was devoted to microscopy analysis.
Morphology and additional elemental composition analysis was carried out on this set of aluminum
substrates using scanning electron microscopy equipped with energy dispersive X-ray
spectroscopy (SEM-EDX) in the Kuiper Imaging cores at the University of Arizona. Secondary





electron (SE) imaging and EDX elemental analysis were performed using a Hitachi S-4800 high
resolution SEM coupled to a Noran system Six X-ray Microanalysis System by Thermo Fisher
Scientific. EDX analysis on individual particles was performed with 30 kV accelerating voltage to
obtain weight percentages of individual elements. SEM-EDX results showed that the background
control aluminum substrate was dominated by Al (88.27%), with minor contributions from Ag
(5.34%), C (4.87%), O (0.79%), Fe (0.67%), and Co (0.05%). Such contributions were manually
subtracted from spectra of individual particles on sample substrates, with the remaining elements
scaled up to hundred percent.  Image processing was conducted with Image J software to measure
particle dimensions and adjust the contrast and brightness of images to provide better visualization.
**2.5 Computational Analysis**

This study reports basic descriptive statistics for chemical concentrations and correlations

between different variables. Statistical significance hereafter corresponds to 95% significance
based on a two-tailed Student's t-test. To complement correlative analysis for identifying sources
of species, positive matrix factorization (PMF) modeling was carried out using the United States
Environmental Protection Agency's (US EPA) PMF version 5. Species considered as "strong"
based on high signal-to-noise ratios (S/N > 1) and those with at least 50% of the concentrations
above the detection limit were used in the PMF modeling (Norris et al., 2014). Data points with
concentrations exceeding the LOD had uncertainty quantified as follows:

$\sigma_{ij} = 0.05 \cdot X_{ij} + LOD_{ij}$ (Equation 1)

where $\sigma_{ij}$, $X_{ij}$, and $LOD_{ij}$ are the uncertainty, concentration, and LOD, respectively, of the $j$[th]
species in the $i$[th] sample (Reff et al., 2007). When concentration data were not available for a
particular stage of a MOUDI set for a species, the geometric mean of the concentrations for that
MOUDI stage and species was applied with uncertainty counted as four times the geometric mean
value (Polissar et al., 1998; Huang et al., 1999).  A 25% extra modeling uncertainty was applied
to account for other sources of errors such as changes in the source profiles and chemical
transformations (Dumanoglu et al., 2014; Norris et al., 2014).  The model was run 20 times with a
randomly chosen starting point for each run.
**3. Results**



## 3.1 Total Mass Concentrations and Charge Balance


The average total mass concentration (± standard deviation) of water-soluble species across

all MOUDI stages (Table 1) during the study period was $8.53 \pm 4.48$ µg m$^{-3}$ (range = 2.7–16.6 µg
m$^{-3}$). The species contributing the most to the total water-soluble mass concentration during the
SWM included SO$_4^{2-}$ (44% ± 6%), NH$_4^+$ (18% ± 5%), NO$_3^-$ (10 ± 3%), Na$^+$ (8 ± 3%), and Cl$^-$ (6%
± 3%). The meteorological parameters from Table 1 best correlated to total water-soluble mass
concentrations were temperature (r = 0.64) and rainfall (r = -0.49). The highest total mass
concentration (set MO13/14 = 16.6 µg m$^{-3}$) occurred during the period with one of the highest
average temperatures (27.8 °C) and second least total rainfall (0.8 mm). Other sampling periods
with high mass concentrations (sets MO7, MO8, and MO12) coincided with the highest
temperature and lowest rainfall observations. High temperatures, and thus more incident solar
radiation, presumably enhanced production of secondary aerosol species via photochemical
reactions as has also been observed in other regions for their respective monsoon season (Youn et
al., 2013). Low rainfall is thought to have been coincident with reduced wet scavenging of aerosol
at the study site as has been demonstrated for other regions such as North America (Tai et al.,
2010) and megacities such as Tehran (Crosbie et al., 2014). However, set MO11 exhibited a very
low concentration even with high temperature and lack of rainfall, which may be due to changes
in the source and transport of aerosol since this sample set coincided with a significant change in
average wind direction (290.2° for MO11 vs. 90.1° – 127.5° for all other MOUDI sets). While the
reported rainfall measurements were taken at MO, inhomogeneous rainfall patterns in the regions
surrounding the Philippines could also contribute to the wet scavenging of PM, thereby lowering
the quantity of transported particles reaching the sample site. Future work will address the
influence of spatiotemporal patterns of precipitation on PM loadings in the Philippines as a point
measurement at an aerosol observing site may be misleading.

On two occasions, two simultaneous MOUDI sets (Sets MO3/MO4 and MO13/MO14)

were collected for the potential to compare different properties that require separate substrates.
The total mass concentrations based on gravimetric analysis of sets MO3 and MO13 were 18.6 µg
m$^{-3}$ and 53.0 µg m$^{-3}$, respectively (Figure 2). Both sets exhibited a dominant concentration mode
between 0.32–0.56 µm and the MO3 set was different in that it exhibited bimodal behavior with a
second peak between 1.8–3.2 µm. The sum of speciated water-soluble species accounted for only
27.8% and 31.3% of the total gravimetric mass of sets MO3 and MO13, respectively, indicative



of significant amounts of water-insoluble species undetected by IC and ICP-QQQ. When adding
the total mass of BC (14.3 µg m$^{-3}$) to the other resolved species from set MO13 (the one time BC
was measured), there was still 22.1 µg m$^{-3}$ of unresolved mass (41.8% of total PM). Most of the
unaccounted mass was for $D_p > 0.32$ µm. The observation of BC accounting for 26.9% of total PM
(14.3 µg m$^{-3}$) is consistent with past work highlighting the significant fraction of BC in the ambient
aerosol of Manila (Kim Oanh et al., 2006; Bautista et al., 2014; Simpas et al., 2014; Kecorius et
al., 2017). However, this fraction of BC is very high compared to measurements during the
monsoon season in other parts of the world. The mass fraction of BC in total suspended PM
(TSPM) was 1.6%/2.2% for the monsoon season in 2013/2014 in Kadapa in southern India, even
though the TSPM measured was comparable to that in Manila (64.9 and 49.9 µg m$^{-3}$, for 2013 and
2014 in Kadapa, respectively) (Begam et al., 2017). Multiple studies during the monsoon season
in a coastal region in southwest India showed BC mass contributions of 1.9 – 5% (Aswini et al.,
2019 and references therein). Airborne measurements around North America and in Asian outflow
revealed that BC accounted for only ~1-2% of PM$_{1.0}$ (Shingler et al., 2016) and ~5-15% of
accumulation mode aerosol mass (Clarke et al., 2004), respectively.

To investigate further about the missing species, a charge balance was carried out for all

MOUDI sets (Table 2) to compare the sum of charges for cations versus anions based on IC
analysis including K from ICP-QQQ analysis (species listed in Section 2.3). The slope of the
charge balances (cations on y-axis) for the cumulative dataset was 1.33 and ranged from 0.89 to
1.41 for the 12 individual MOUDI sets that had IC and ICP-QQQ analysis conducted on them.
Eleven of the 12 sets exhibited slopes above unity indicating that there was a deficit in the amount
of anions detected, which presumably included species such as carbonate and various organics. To
further determine if there were especially large anion or cation deficits in specific size ranges,
slopes are also reported for 0.056–1 µm and > 1 µm. There were no obvious differences other than
two MOUDI sets exhibited slopes below 1.0 for the smaller diameter range (0.056–1 µm) while
all slopes exceeded unity for > 1 µm.

## 3.2 Mass Size Distributions and Morphology

### 3.2.1 Black Carbon

The size-resolved nature of BC has not been characterized in Manila and MOUDI set

MO13 offered a view into its mass size distribution (Figure 3a). There was a pronounced peak



between 0.18–0.32 µm (5.0 µg m$^{-3}$), which is evident visually in the substrate's color when
compared to all other stages of that MOUDI set (Figure 3b). This observed peak in the mass size
distribution of BC is similar to previous studies of the outflow of East Asian countries (Shiraiwa
et al., 2008), biomass burning and urban emissions in Texas (Schwarz et al., 2008), measurements
in the Finnish Arctic (Raatikainen et al., 2015), and airborne measurements over Europe
(Reddington et al., 2013). In contrast, measurements in Uji, Japan showed a bimodal size
distribution for the mass concentration of BC in the submicrometer range (Hitzenberger and
Tohno, 2001). In the present study, there were significant amounts of BC extending to as low as
the 0.056-0.1 µm MOUDI stage (0.28 µg m$^{-3}$) and extending up in the supermicrometer range with
up to 0.25 µg m$^{-3}$ measured between 1.8–3.2 µm.  Remarkably, BC accounted for approximately
78.1% (51.8%) by mass of the total PM in the range of 0.10 – 0.18 µm (0.18 – 0.32 µm). For
comparison, the mass percent contribution of BC measured in the megacity of Nanjing, China was
3.3% (1.6%) at 0.12 (0.08) µm (Ma et al., 2017).  Based on visual inspection of color on all
MOUDI sets, MO13 appears to be representative of the other sets based on the relative intensity
of the color black on substrates with different cutpoint diameters (Figure 3b); the 0.18–0.32 µm
substrate always was the most black, with varying degrees of blackness extending consistently into
the supermicrometer stages.

Microscopy analysis revealed evidence of non-spherical particles in each MOUDI stage

below 1 µm (Figure 4), which is significant as the common assumption theoretically is that
submicrometer particles are typically spherical (e.g., Mielonen et al., 2011). Errors in this
assumption impact numerical modeling results and interpretation of remote sensing data for
aerosols (e.g., Kahnert et al., 2005) owing to incorrect calculations of parameters such as single
scattering albedo, asymmetry parameter, and extinction efficiency (e.g., Mishra et al., 2015). Some
studies have noted that submicrometer particles could be composed of an agglomeration of small
spherical particles originally formed through gas-to-particle conversion processes (Almeida et al.,
2019), which could potentially explain the appearance for some of the observed particles in Figure
4. Since only single particles were examined that may not be fully representative of all particles
on a particular MOUDI substrate, it is noteworthy that all five particles shown between 0.056 – 1
µm were irregularly shaped with signs of both multi-layering and constituents adhered to one
another. The images show that a potentially important source of BC in the area could be soot
aggregates, which are formed by a vaporization-condensation process during combustion often



associated with vehicular exhaust (e.g., Chen et al., 2006; Chithra and Nagendra, 2013; Wu et al.,
2017). Kecorius et al. (2017) projected that 94% of total roadside refractory PM in the same study
region was linked to *jeepneys*, with number concentration modes at 20 and 80 nm. They associated
the larger mode with soot agglomerates, which is consistent with the smallest MOUDI size range
examined here (0.056-0.1 μm; Figure 4b) exhibiting signs of agglomeration.

The total BC mass concentration integrated across all stages of MOUDI set MO13 (14.3

μg m$^{-3}$) was remarkably high in contrast to BC levels measured via either filters, aethalometers, or
single particle soot photometers in most other urban regions of the world (Metcalf et al., 2012 and
references therein): Los Angeles Basin (airborne: 0.002–0.53 μg m$^{-3}$), Atlanta, Georgia (ground:
0.5–3.0 μg m$^{-3}$), Mexico City (airborne: 0.276–1.1 μg m$^{-3}$), Sapporo, Japan (ground: 2.3–8.0 μg
m$^{-3}$), Beijing, China (ground: 6.3–11.1 μg m$^{-3}$), Bangalor, India (ground: 0.4–10.2 μg m$^{-3}$), Paris,
France (ground: 7.9 μg m$^{-3}$), Dushanbe, Russia (ground: 4–20 μg m$^{-3}$), Po Valley, Italy (ground:
0.5–1.5 μg m$^{-3}$), Thessaloniki, Greece (ground: 3.3–8.9 μg m$^{-3}$). This is intriguing in light of
extensive precipitation, and thus wet scavenging of PM, during the study period, which is offset
by enormous anthropogenic emissions in the region such as by powered vehicles like the *jeepneys*
that are notorious for BC exhaust (Kecorius et al., 2017).

A possible explanation for the large contribution of BC to PM, and the persistence of PM

after rain events (Kim Oanh et al., 2006), is that the BC is not efficiently scavenged by precipitating
rain drops. Small particles enter rain drops via diffusion whereas large particles enter via
impaction. However, particles with a diameter in the range of 0.1–1 μm (known as the Greenfield
gap) are too large to diffuse efficiently and too small to impact, and are therefore not efficiently
scavenged (Seinfeld and Pandis, 2016). Absorption spectroscopy of set MO13 (Figure 2b) reveals
that 95% of the BC mass is concentrated in the Greenfield gap, and thus the removal of BC due to
precipitation is inefficient. The Greenfield gap contains $62 \pm 11\%$ of the total mass (calculated for
MO3/MO13) and $65 \pm 10\%$ of the water-soluble mass (calculated for the other 12 MO sets).

### 3.2.2 Water-Soluble Ions

There were two characteristic mass size distribution profiles for the water-soluble ions

speciated by IC depending on whether the species were secondarily produced via gas-to-particle
conversion or associated with primarily emitted supermicrometer particles. The average IC species
mass concentration profile across all MOUDI sets is shown in Figure 5. Secondarily produced



species exhibited a mass concentration mode between 0.32–0.56 μm, including common inorganic
species ($SO_4^{2-}$, $NH_4^+$), MSA, amines (DMA, TMA+DEA), and a suite of organic acids, such as
oxalate, phthalate, succinate, and adipate, produced via precursor volatile organic compounds
(VOCs). Two organic acids with peaks in other size ranges included maleate (0.56–1 μm) and
pyruvate (0.1–0.18 μm). Sources of the inorganics are well documented with $SO_4^{2-}$ and $NH_4^+$
produced by precursor vapors $SO_2$ and $NH_3$, respectively, with ocean-emitted dimethylsulfide
(DMS) as an additional precursor to $SO_4^{2-}$ and the primary precursor to MSA.

Precursors leading to secondarily produced alkyl amines such as DMA, TMA, and DEA

likely originated from a combination of industrial activity, marine emissions, biomass burning,
vehicular activity, sewage treatment, waste incineration, and the food industry (e.g., Facchini et
al., 2008; Sorooshian et al., 2009; Ge et al., 2011; VandenBoer et al., 2011); another key source of
these species, animal husbandry (Mosier et al., 1973; Schade and Crutzen, 1995; Sorooshian et al.,
2008), was ruled out owing to a scarcity of such activity in the study region. Secondarily produced
amine salts likely were formed with $SO_4^{2-}$ as the chief anion owing to its much higher
concentrations relative to $NO_3^-$ or organic acids. Dimethylamine was the most abundant amine
similar to other marine (Muller et al., 2009) and urban regions (Youn et al., 2015); the average
concentration of DMA integrated over all MOUDI stages for all sample sets was 62.2 ng m$^{-3}$ in
contrast to 29.8 ng m$^{-3}$ for TMA+DEA. For reference, the other key cation ($NH_4^+$) participating in
salt formation with acids such as $H_2SO_4$ and $HNO_3$ was expectedly much more abundant (1.64 μg
m$^{-3}$). With regard to the competitive uptake of DMA versus $NH_3$ in particles, the molar ratio of
DMA:$NH_4^+$ exhibited a unimodal profile between 0.1–1.8 μm with a peak of 0.022 between 0.32–
0.56 μm and the lowest values at the tails (0.004 between  0.1–0.18 and 1–1.8 μm); DMA was not
above detection limits for either $D_p < 0.1$ μm or $D_p > 1.8$ μm. The molar ratios observed were
consistent with values measured in urban air of Tucson, Arizona and coastal air in Marina,
California (0–0.04; Youn et al., 2015) and near the lower end of the range measured in rural and
urban air masses sampled near Toronto (0.005–0.2: VandenBoer et al., 2011).

The most abundant organic acid was oxalate (195 ± 144 ng m$^{-3}$), followed by succinate (21

± 41 ng m$^{-3}$), phthalate (19 ± 25 ng m$^{-3}$), maleate (17 ± 15 ng m$^{-3}$), and adipate (5 ± 8 ng m$^{-3}$). The
observation of mass concentrations increasing with decreasing carbon number for dicarboxylic
acids (i.e., oxalate > succinate > adipate) is consistent with many past studies for other regions as
larger chain acids undergo oxidative decay to eventually form oxalate (e.g., Kawamura and



Ikushima, 1993; Kawamura and Sakaguchi, 1999; Sorooshian et al., 2007). Maleate is an
unsaturated dicarboxylic acid emitted from gas and diesel engines (Rogge et al., 1993) and a
product from the photo-oxidation of benzene (Kawamura and Ikushima, 1993). The aromatic
dicarboxylic acid phthalate is a known photo-oxidation product of naphthalene and stems largely
from plastic processing and fuel combustion (Fraser et al., 2003; Kautzman et al., 2010; Fu et al.,
2012; Kleindienst et al., 2012). The oxidation product (MSA) of ocean-derived DMS exhibited an
overall average concentration of $11 \pm 7$ ng m$^{-3}$, which is near the lower end of the range of levels
reported in other coastal and marine environments (from undetected up to ~200 ng m$^{-3}$) (e.g.,
Saltzman et al., 1983, 1986; Berresheim 1987; Watts et al., 1987; Burgermeister and Georgii,
1991; Sorooshian et al., 2015; Xu and Gao, 2015).
Water-soluble species exhibiting a peak in the supermicrometer range, usually between
1.8–5.6 μm, include those with known affiliations with sea salt ($Na^+$, $Cl^-$, K+, $Mg^{2+}$) and crustal
materials such as dust ($Ca^{2+}$). Nitrate peaked between 1.8-3.2 μm, and was best correlated with
$Na^+$ and $Mg^{2+}$, suggestive of $HNO_3$ partitioning to sea salt as has been observed in other coastal
regions (e.g., Prabhakar et al., 2014a). There was very little $NO_3^-$ in the submicrometer range (0.05
$\pm$ 0.04 μg m$^{-3}$) in contrast to supermicrometer sizes ($0.78 \pm 0.47$ μg m$^{-3}$). More submicrometer
$NO_3^-$ in the form of $NH_4NO_3$ would be expected if there was an excess of $NH_3$ after neutralizing
$SO_4^{2-}$. The mean ammonium-to-sulfate molar ratio for submicrometer sizes was $2.32 \pm 0.52$ (range:
1.11 – 2.78), with full neutralization of $SO_4^{2-}$ in 10 of 12 MOUDI sets. Thus, there was a non-
negligible excess in $NH_3$ that presumably participated in salt formation with $HNO_3$ and organic
species. The significant levels of $NO_3^-$ in the same mode as $Na^+$ and $Cl^-$ contributed to the
significant $Cl^-$ depletion observed, as the mean $Cl^-$:$Na^+$ mass ratio between 1-10 μm (i.e., range of
peak sea salt influence) was $0.81 \pm 0.28$, which is much lower than the ratio for pure sea salt (1.81)
(Martens et al., 1973). The subject of $Cl^-$ depletion in this region will be investigated more
thoroughly in subsequent work.
Figure 6 shows SEM images of representative single particles in each supermicrometer
stage. As would be expected for sea salt and crustal material, most of the particles shown are not
spherical. Interestingly, only the particle shown between 1–1.8 μm was close to being spherical.
Its composition based on EDX analysis was accounted for mostly by carbon (93.7%) with lower
amounts of oxygen (5.8%) and Fe (0.5%). Sea salt particles were found in the next two stages
owing to the highest combined weight percentages of $Na^+$ and $Cl^-$ based on EDX analysis: 1.8–3.2



µm = 36.9%; 3.2–5.6 µm = 46.9%. The salt particles are not necessarily cubical but more rounded
with signs of agglomeration. These two particles were the only ones among the 11 MOUDI stages
exhibiting an EDX signal for S, with contributions amounting to ~2% in each particle. This may
be linked to natural $SO_4^{2-}$ existing in sea salt particles. Also, the particle between 3.2–5.6 µm
contained a trace amount of Sc (1%). The largest three particles (≥ 5.6 µm) were expectedly
irregularly shaped with both sharp and rounded edges, comprised mostly of oxygen, Al, Fe, and
Ca based on EDX analysis.
**3.2.3 Water-Soluble Elements**
Averaged data across all MOUDI sets reveal that ICP-QQQ elements exhibited a variety
of mass concentration profiles ranging from a distinct mode in either the sub- or supermicrometer
range to having multiple modes below and above 1 µm (averages across all MOUDI sets shown
in Figure 7). There were several elements with only one distinct peak, being in one of the two
stages between 0.18-1.0 µm, including As, Cd, Co, Cr, Cs, Cu, Hf, Mn, Mo, Ni, Rb, Se, Sn, Tl, V,
Pb, and Zn. In contrast, the following elements exhibited only one distinct peak in the
supermicrometer range: Al, Ba, P, Sr, Ti, Y, and Zr. The rest of the elements exhibited more
complex behavior with two distinct peaks in the sub- and supermicrometer range (Ag, Fe, Nb).
The following section discusses relationships between all of the ions and elements with a view
towards identifying characteristic sources.
**3.3 Characteristic Sources and Species Relationships**
A combination of PMF and correlation analysis helped identify clusters of closely related
species stemming from distinct sources. The final PMF solution, based on five groups of species
(Figure 8), passed criteria associated with being physically valid and the close proximity of the
calculated ratio of $Q_{true}$:$Q_{expected}$ (1.2) to 1.0. There was a high coefficient of variation between
measured and predicted mass concentration when summing up all species for each MOUDI stage
($r^2$ = 0.79; sample size, n = 132), which added confidence in relying on the PMF model for source
apportionment of PM. The five distinct clusters were named for their most plausible sources based
on the species included in the groupings, with their overall contributions to the total mass based
on PMF analysis shown in parenthesis (Table 3): Aged/Transported (48.0%), Sea Salt (22.5%),
Combustion (18.7%), Vehicular/Resuspended Dust (5.6%), and Waste Processing (5.1%). For



reference, a previous study near the northwestern edge of the Philippines identified six source
factors for PM$_{2.5}$ that are fairly similar to those here (Bagtasa et al., 2018): sea salt, resuspended
fine dust, local solid waste burning, and long range transport of (i) industrial emissions, (ii) solid
waste burning, and (iii) secondary sulfate. Each of our five groupings will be discussed in detail
below in decreasing order of contribution to total measured mass concentrations.

**3.3.1 Aged/Transported Aerosol**

Although not due to one individual source, there was a distinct PMF factor that included
species commonly produced via gas-to-particle conversion processes (NH$_4^+$, SO$_4^{2-}$, MSA,
oxalate). Correlation analysis (Table 4) also pointed to a large cluster of species significantly
related to each other, including the aforementioned ions and a suite of other organic acids
(phthalate, succinate, adipate), MSA, and DMA. The latter three inorganic and organic acid ions
exhibited significant correlations with each other (r $\geq$ 0.68), but also with several elements (r $\geq$
0.36: K, V, Rb, Cs, Sn), which were likely co-emitted with the precursor vapors of the secondarily
produced ions. Although BC concentrations were quantified from set MO13, their
interrelationships with water-soluble ions from simultaneously collected set MO14 are
representative for other sets. The results showed that BC was significantly correlated (r: 0.61-0.92)
with 15 species, including those mentioned above (owing to co-emission) and also a few elements
that were found via PMF to be stronger contributors to the Combustion source discussed in Section
3.3.3 (Ni, Cu, As, Se, Cd, Tl, Pb).
This PMF source factor is referred to as Aged/Transported owing to it characteristic species
being linked to sources distant from the sample site. Examples include MSA and DMA being
secondarily produced from ocean-derived gaseous emissions (e.g., Sorooshian et al., 2009), and K
stemming from biomass burning emissions from upwind regions such as Sumatra and Borneo
(Xian et al., 2013). Previous studies (Reid et al., 2012; Wang et al., 2013) have shown that
phenomena such as SWM and El-Nino events not only influence biomass burning activities in the
Malay Peninsula but also impact the transport and distribution of emissions in the study region.
For instance, Reid et al. (2016b) showed that enhancement in monsoonal flow facilitates the
advection of biomass burning and anthropogenic emissions to the Philippines from Sumatra and
Borneo. Subsequent work will investigate more deeply the impact of biomass burning from those
upwind regions on the sample site during the SWM.



While NH$_4^+$ and SO$_4^{2-}$ require time for production owing to being secondarily produced
from precursor vapors (i.e., SO$_2$, NH$_3$), oxalate is the smallest dicarboxylic acid and requires
lengthier chemistry pathways for its production and thus is more likely produced in instances of
aerosol transport and aging (e.g., Wonaschuetz et al., 2012; Ervens et al., 2018). The various
elements associated with this cluster are co-emitted with the precursors to the aforementioned ions
and are linked to a variety of sources: metallurgical processes (Anderson et al., 1988; Csavina et
al., 2011; Youn et al., 2016), fuel combustion (Nriagu, 1989; Allen et al., 2001; Shafer et al., 2012;
Rocha and Correa, 2018), residual oil combustion (Watson et al., 2004), biomass burning (Maudlin
et al., 2015), marine and terrestrial biogenic emissions (Sorooshian et al., 2015), and plastics
processing (Fraser et al., 2003). In addition, there is extensive ship traffic in the general study
region, which is a major source of species in this cluster of species, particularly V and SO$_4^{2-}$ (e.g.,
Murphy et al., 2009; Coggon et al., 2012).
PMF analysis suggested that the Aged/Transported factor contributed 48.0% to the total
water-soluble mass budget during the study period. Most of the contribution resided in the
submicrometer range (68.9%) unlike the supermicrometer range (18.6%), which is consistent with
the overall mass size distribution of total PM peaking in the submicrometer range (Figure 2). The
reconstructed mass size distribution for this PMF source factor shows the dominance of the mass
in the submicrometer range with a peak between 0.32–0.56 µm (Figure 9). The correlation matrices
for the sub- and supermicrometer size ranges also show that the correlations between the species
most prominent in the Aged/Transported category are stronger for the former size range (Tables
S2-S3). The contribution of this PMF factor to the supermicrometer range is likely associated with
species secondarily produced on coarse aerosol such as dust and sea salt. This is evident in the
individual species mass size distributions where there is a dominant submicrometer mode but also
non-negligible mass above 1 µm.
Even though the PM in a heavily populated urban region, such as Metro Manila, is typically
thought to be dominated by local sources of aerosols, the current PMF results show that the largest
contributions to water-soluble aerosol mass are from Aged/Transported pollution. This finding is
contrary to the expectation that (a) the signal of transported aerosols would be lost in the noise of
locally-produced aerosols, and (b) the removal of aerosols over the ocean surrounding the
Philippines by processes such as wet scavenging would significantly reduce the contribution of
transported aerosols. Even though other cities may have different pollution signatures, varying in



pollutant type and amount, this phenomenon of Aged/Transported pollution forming a significant
portion of the water-soluble mass may be applicable to other cities, especially those in Southeast
Asia.
**3.3.2 Sea Salt**

As the MO sampling site is approximately 13 km from the nearest shoreline (Figure 1a)

and downwind of Manila Bay in the SWM season, there was a great potential for marine emissions
to impact the samples. There were several species with similar mass size distributions (mode: 1.8–
5.6 μm) and highly correlated total mass concentrations ($r \geq 0.51$) that are linked to sea salt: $Cl^-$,
$Na^+$, $Ca^{2+}$, $Mg^{2+}$, Ba, and Sr. The correlations between these species were stronger when examining
just the supermicrometer range as compared to the submicrometer range (Tables S2-S3). The
majority of these species was used in PMF analysis and formed a distinct cluster amounting to
22.0% of the total study period's mass budget. This source contributed only 0.6% to the
submicrometer mass concentration but 53.5% for the supermicrometer size range. The
reconstructed mass size distribution for this source factor is shifted farthest to the larger diameters
as compared to the other four sources with a peak between 1.8-3.2 μm (Figure 9).

It is noteworthy that this factor has the highest share of $NO_3^-$ among all identified sources.

This result is consistent with mass size distributions shown in Figure 5 in which $NO_3^-$ peaks in the
supermicrometer range similar to sea salt constituents (e.g., $Na^+$ and $Cl^-$). Although sea salt
particles naturally contain $NO_3^-$ (Seinfeld and Pandis, 2016) (mass ratio of $NO_3^-:Na^+ = 9.8 \times 10^{-8}$
$- 6.5 \times 10^{-5}$), the extremely high ratio of $NO_3^-:Na^+$ (mass ratio ~1.8) suggests that only a negligible
portion of $NO_3^-$ in this factor originated from primary sea salt particles. Thus, the majority of $NO_3^-$
is most likely due to $HNO_3$ partitioning to existing sea salt particles (e.g., Fitzgerald, 1991; Allen
et al., 1996; Dasgupta et al., 2007; Maudlin et al., 2015). In addition, the $Cl^-:Na^+$ mass ratio in this
profile (0.65) is smaller than that in sea salt particles (1.81), indicating high $Cl^-$ depletion mainly
due to reactions of $HNO_3$ with NaCl (Ro et al., 2001; Yao et al., 2003; Braun et al., 2017).
Moreover, elevated loadings of trace elements (e.g., Ba, Cu, Zn, and Co) could be linked to mixing
of marine emissions with urban sources (e.g., vehicle and industrial emissions) during their
transport inland to the sampling site (Roth and Okada, 1998). This process of aging is consistent
with the observed morphology of the sea salt particles in this study, revealing non-cubical shapes
that are rounded owing to the likely addition of acidic species such as $HNO_3$ (Figure 6).



### 3.3.3 Combustion

There are numerous sources of combustion in the study region including a variety of mobile sources (e.g., cars, utility vehicles, trucks, buses, motorcycles) and stationary sources (e.g., power stations, cement works, oil refineries, boiler stations, utility boilers). Consequently, the next highest contributor to total mass during the study period according to PMF (18.7%) was the cluster of species including Ni, As, Co, P, Mo, and Cr, which is defined as the Combustion factor. These species have been reported to be rich in particles emitted from combustion of fossil fuel and residual oil (Linak and Miller, 2000; Allen et al., 2001; Wasson et al., 2005; Mahowald et al., 2008; Mooibroek et al., 2011; Prabhakar et al., 2014b). Although not included in PMF analysis, other species significantly correlated with the previous ones include maleate and Ag, which also stem from fuel combustion (Kawamura and Kaplan, 1987; Lin et al., 2005; Sorooshian et al., 2007). Ag specifically is an element in waste incinerator fly ash (Buchholz and Landsberger, 1993; Tsakalou et al., 2018) and its strong correlation with Co ($r = 0.85$) and Mo ($r = 0.64$) provides support for this source factor being linked to combustion processes. Maleate is commonly found in engine exhaust (Kawamura and Kaplan, 1987), while Cr is a tracer for power plant emissions (Singh et al., 2002; Behera et al., 2015). Of all species examined in this study, BC was best correlated with As ($r = 0.92$), while its correlation with Ni ($r = 0.85$) was among the highest.

As the elements in this cluster peaked in concentration in the submicrometer mode, the weight percentage of this factor is more than double below 1 µm (23.9%) as compared to above 1 µm (11.3%). The reconstructed mass size distribution for this source factor peaks between 0.18–0.32 µm, which is smaller than the modal diameter range for the Aged/Transported source factor (0.32–0.56 µm) likely owing to closer sources and thus less time for growth to occur via condensation and coagulation.

### 3.3.4 Vehicular/Resuspended Dust

The next PMF source factor contains chemical signatures of dust because of high contributions to Al, Ti, Ca, and Fe. These crustal elements are strongly related to resuspension of dust by traffic and construction activities (Singh et al., 2002; Harrison et al., 2011). Other elements that were prominent in this factor included Zr, Y, Mn, Cr, and Ba, which are associated with tire and brake wear (Adachi and Tainosho, 2004; Gietl et al., 2010; Song and Gao, 2011; Harrison et al., 2012; Vossler et al., 2016), although some of them can be linked to the exhaust as well (e.g.,




Lin et al., 2005; Song and Gao, 2011). This source is named Vehicular/Resuspended Dust and
contributed 5.6% to the total study period's mass concentrations.
The weight percentage contribution of this factor was much higher for the supermicrometer
range (11.3%) as compared to the submicrometer range (1.5%), which is consistent with the Sea
Salt source factor owing to similar mass size distributions of the individual species associated with
the two source categories (Figures 5 and 7). Additional species correlated significantly with the
crustal species included Hf and Nb, which also exhibited mass peaks between 1.8–3.2 μm. The
reconstructed mass size distribution for this source factor is similar to that of Sea Salt in that there
is a peak between 1.8–3.2 μm, but there is less of a unimodal profile owing to what appears to be
a secondary mode between 0.56–1.0 μm (Figure 9), which could be linked to some of the non-dust
components of vehicular emissions.

### 586  3.3.5 Waste Processing

The final PMF source factor, contributing the least overall to total mass (5.1%), featured
Zn, Cd, Pb, Mn, and Cu as its main components. These species are linked to waste processing,
including especially electronic waste (e-waste) and battery burning and recycling (Gullett et al.,
2007; Iijima et al., 2007), which was previously reported for Manila (Pabroa et al., 2011). The
latter study reported that although there are a few licensed operations for battery recycling, there
are numerous unregulated cottage melters across Manila that regularly melt metal from batteries
and discard the waste freely. Fujimori et al. (2012) additionally showed that e-waste recycling led
to emissions of the following elements (in agreement with this PMF cluster) around Metro Manila:
Ni, Cu, Pb, Zn, Cd, Ag, in, As, Co, Fe, and Mn.
This was the only PMF factor exhibiting comparable weight percentages both below
(5.1%) and above 1 μm (5.3%). This is reflected in the mass size distributions of the species
included in this cluster being fairly uniformly distributed below and above 1 μm. This is also
demonstrated in the reconstructed mass size distribution of this source factor as it clearly exhibits
a mode between the other four sources (0.56–1.0 μm) and is the broadest mode (Figure 9). The
explanation for this is likely rooted in the diversity of sources contained within this source profile
that lead to different sizes of particles. Examples of such sources include processing of different
types of waste at varying temperatures and through various processes (e.g., burning, melting,
grinding) (Keshtkar and Ashbaugh, 2007),



## 4. Conclusions

This study used various analytical techniques (gravimetry, IC, ICP-QQQ, black carbon spectroscopy, and microscopy), meteorological data, and a source apportionment model (PMF) to characterize the sources, chemical composition, and morphology of size-resolved ambient PM in Metro Manila, Philippines during the SWM season of 2018. The main results of this study include the following:

- The total mass concentrations were measured on two occasions and were 18.6 μg m⁻³ and 53.0 μg m⁻³. Water-soluble mass concentrations were measured on 12 occasions and were on average $8.53 \pm 4.48$ μg m⁻³ (range = 2.7–16.6 μg m⁻³). Simultaneous measurements of total, water-soluble, and BC mass revealed a composition of 26.9% BC, 31.3% water-soluble components, and 41.8% unaccounted mass.

- Size-resolved BC mass concentration was measured on one occasion, with the mass sum of all MOUDI stages reaching 14.3 μg m⁻³. Most of the BC mass (95%) was contained in the 0.1–1 μm range (i.e., the Greenfield gap) where wet scavenging by rain is inefficient. The measured BC peaked in the size range of 0.18 – 0.32 μm and accounted for 51.8% of the measured PM for that stage. In the range of 0.10 – 0.18 μm, the mass percent contribution of BC to the measured PM was 78.1%.

- Most of the total mass resided in the submicrometer mode (0.32–0.56 μm); however, one MOUDI set revealed an additional supermicrometer mode (1.8–3.2 μm). Water-soluble species that peaked in the submicrometer mode were associated with secondarily produced species, including inorganic acids, amines, MSA, and organic acids. Water-soluble species that peaked in the supermicrometer mode were associated with sea salt and crustal material. Most of the unaccounted mass was for $D_p > 0.32$ μm.

- The most abundant water-soluble species was $SO_4^{2-}$ (44% ± 6%), followed by $NH_4^+$ (18% ± 5%), $NO_3^-$ (10 ± 3%), $Na^+$ (8 ± 3%), and $Cl^-$ (6% ± 3%). Correlation analysis revealed that total water-soluble mass was most correlated with temperature (r = 0.64) and rainfall accumulation (r = -0.49) among meteorological factors considered, although other factors were likely influential such as wind direction and speed.

- Regardless of particle size, the majority of single particles examined with SEM-EDX were non-spherical with evidence of agglomeration.



- PMF analysis suggested that there were five factors influencing the water-soluble fraction of PM collected at the sampling site. These factors, their contribution to total water-soluble mass, and the main species that permit them to be linked to a physical source are as follows: Aged/Transported (48.0%; $NH_4^+$, $SO_4^{2-}$, MSA, oxalate), Sea Salt (22.5%; $Cl^-$, $NO_3^-$, $Ca^{2+}$, $Na^+$, $Mg^{2+}$, Ba, Sr), Combustion (18.7%; Ni, As, Co, P, Mo, Cr), Vehicular/Resuspended Dust (5.6%; Al, Ti, Fe), and Waste Processing (5.1%; Zn, Cd, Pb, Mn, Cu). The dominant contribution of Aged/Transported aerosols to water-soluble mass contradicts two expectations: (i) locally-produced sources in polluted cities should drown out the signal of transported aerosols, and (ii) the signal of transported aerosols should be significantly reduced due to scavenging processes upwind of the measurement site.

Although the current study focuses exclusively on the SWM season in Metro Manila, results of this study are applicable to the study of aerosol impacts on Southeast Asia and other regions. First, the significant presence of Aged/Transported aerosols in Metro Manila indicates that PM in the region has the ability to travel long distances during the SWM season, despite the typical assumption that wet scavenging effectively removes most of the particles. Characterization of aerosols in Metro Manila is therefore important for better understanding the impacts that local emissions will have on locations downwind of Metro Manila, including other populated cities in Southeast and East Asia. Transport of pollution and decreased wet scavenging during the SWM season may become increasingly important as studies have shown a decrease in SWM rainfall and increase in the number of no-rain days during the SWM season in the western Philippines in recent decades (e.g., Cruz et al., 2013).

Second, Southeast Asia has been named "one of the most hostile environments on the planet for aerosol remote sensing" (Reid et al., 2013). Therefore, space-based remote sensing of aerosol characteristics, such as retrievals of aerosol optical depth (AOD), in this region are difficult. In situ measurements are critical for characterization of PM in this region, especially during seasons such as the SWM when clouds are especially prevalent and remote-sensing retrievals dependent on clear-sky conditions are lacking.

Third, this study provides a valuable dataset to compare to other regions impacted by monsoons where the impacts of enhanced moisture and rainfall on size-resolved composition are not well understood. As aqueous processing results in enhanced production of water-soluble

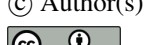



species (e.g., sulfate, organic acids), it is noteworthy for this monsoonal region that the water-soluble fraction remains low relative to BC and other insoluble components. This has major implications for the hygroscopicity of the regional PM.

Finally, the results of this study will be used to inform future sampling campaigns in this region, including CAMP$^2$Ex planned for the SWM season of 2019 based in the Philippines. As the current MOUDI sampling campaign at MO is expected to extend for a full year, future work will focus on changes in aerosol characteristics and sources on a seasonal basis.

*Data availability:* All data used in this work are available upon request.

*Author Contribution:* MTC, MOC, JBS, ABM, CS, and AS designed the experiments and all co-authors carried out some aspect of the data collection. MTC, RAB, CS, LM, HD, and AS conducted data analysis and interpretation. MTC and AS prepared the manuscript with contributions from all co-authors.

*Competing interests:* The authors declare that they have no conflict of interest.

*Acknowledgements:* This research was funded by NASA grant 80NSSC18K0148. M. T. Cruz acknowledges support from the Philippine Department of Science and Technology's ASTHRD Program. R. A. Braun acknowledges support from the ARCS Foundation. A. B. MacDonald acknowledges support from the Mexican National Council for Science and Technology (CONACYT). We acknowledge Agilent Technologies for their support and Shane Snyder's laboratories for ICP-QQQ data.

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

Contributions of Brake Dust, Tire Wear, and Resuspension to Nonexhaust Traffic Particles
Derived from Atmospheric Measurements, Environ Sci Technol, 46, 6523-6529,
10.1021/es300894r, 2012.
Hitzenberger, R., and Tohno, S.: Comparison of black carbon (BC) aerosols in two urban areas –
concentrations and size distributions, Atmospheric Environment, 35, 2153-2167,
https://doi.org/10.1016/S1352-2310(00)00480-5, 2001.
Hopke, P. K., Cohen, D. D., Begum, B. A., Biswas, S. K., Ni, B., Pandit, G. G., Santoso, M.,
Chung, Y. S., Davy, P., Markwitz, A., Waheed, S., Siddique, N., Santos, F. L., Pabroa, P. C. B.,
Seneviratne, M. C. S., Wimolwattanapun, W., Bunprapob, S., Vuong, T. B., Duy Hien, P. and
Markowicz, A.: Urban air quality in the Asian region, Sci. Total Environ., 404(1), 103–112,
doi:10.1016/j.scitotenv.2008.05.039, 2008.
Huang, S. L., Rahn, K. A., and Arimoto, R.: Testing and optimizing two factor-analysis techniques
on aerosol at Narragansett, Rhode Island, Atmos Environ, 33, 2169-2185, Doi 10.1016/S1352-
875    2310(98)00324-0, 1999.
Iijima, A., Sato, K., Yano, K., Tago, H., Kato, M., Kimura, H., and Furuta, N.: Particle size and
composition distribution analysis of automotive brake abrasion dusts for the evaluation of
antimony     sources     of     airborne     particulate     matter,     Atmos     Environ,     41,     4908-4919,
10.1016/j.atmosenv.2007.02.005, 2007.
Kahnert, M., Nousiainen, T., and Veihelmann, B.: Spherical and spheroidal model particles as an
error source in aerosol climate forcing and radiance computations: A case study for feldspar
aerosols, J Geophys Res-Atmos, 110, Artn D18s13, 10.1029/2004jd005558, 2005.




Kautzman, K. E., Surratt, J. D., Chan, M. N., Chan, A. W. H., Hersey, S. P., Chhabra, P. S.,
Dalleska, N. F., Wennberg, P. O., Flagan, R. C., and Seinfeld, J. H.: Chemical Composition of
Gas- and Aerosol-Phase Products from the Photooxidation of Naphthalene, J Phys Chem A, 114,
913-934, 10.1021/jp908530s, 2010.
Kawamura, K., and Ikushima, K.: Seasonal-Changes in the Distribution of Dicarboxylic-Acids in
the Urban Atmosphere, Environ Sci Technol, 27, 2227-2235, DOI 10.1021/es00047a033, 1993.
Kawamura, K., and Kaplan, I. R.: Motor Exhaust Emissions as a Primary Source for Dicarboxylic-
Acids  in  Los-Angeles  Ambient  Air,  Environ  Sci  Technol,  21,  105-110,  DOI
10.1021/es00155a014, 1987.
Kawamura, K., and Sakaguchi, F.: Molecular distributions of water soluble dicarboxylic acids in
marine aerosols over the Pacific Ocean including tropics, J Geophys Res-Atmos, 104, 3501-3509,
Doi 10.1029/1998jd100041, 1999.
Kecorius, S., Madueño, L., Löndahl, J., Vallar, E., Galvez, M. C., Idolor, L. F., Gonzaga-
Cayetano, M., Müller, T., Birmili, W., and Wiedensohler, A.: Respiratory tract deposition of
inhaled roadside ultrafine refractory particles in a polluted megacity of South-East Asia, Science
of The Total Environment, 663, 265-274, https://doi.org/10.1016/j.scitotenv.2019.01.338, 2019.
Kecorius, S., Madueno, L., Vallar, E., Alas, H., Betito, G., Birmili, W., Cambaliza, M. O., Catipay,
G., Gonzaga-Cayetano, M., Galvez, M. C., Lorenzo, G., Muller, T., Simpas, J. B., Tamayo, E. G.,
and Wiedensohler, A.: Aerosol particle mixing state, refractory particle number size distributions
and emission factors in a polluted urban environment: Case study of Metro Manila, Philippines,
Atmos Environ, 170, 169-183, 10.1016/j.atmosenv.2017.09.037, 2017.
Keshtkar, H., and Ashbaugh, L. L.: Size distribution of polycyclic aromatic hydrocarbon
particulate emission factors from agricultural burning, Atmos Environ, 41, 2729-2739,
10.1016/j.atmosenv.2006.11.043, 2007.
Kim Oanh, N. T., Upadhyay, N., Zhuang, Y. H., Hao, Z. P., Murthy, D. V. S., Lestari, P., Villarin,
J. T., Chengchua, K., Co, H. X., Dung, N. T. and Lindgren, E. S.: Particulate air pollution in six
Asian cities: Spatial and temporal distributions, and associated sources, Atmos. Environ., 40(18),
3367–3380, doi:10.1016/j.atmosenv.2006.01.050, 2006.
Kleindienst, T. E., Jaoui, M., Lewandowski, M., Offenberg, J. H., and Docherty, K. S.: The
formation of SOA and chemical tracer compounds from the photooxidation of naphthalene and its
methyl analogs in the presence and absence of nitrogen oxides, Atmos Chem Phys, 12, 8711-8726,
10.5194/acp-12-8711-2012, 2012.
Liao, H., Chen, W. T., and Seinfeld, J. H.: Role of climate change in global predictions of future
tropospheric ozone and aerosols, J Geophys Res-Atmos, 111, Artn D12304,
10.1029/2005jd006852, 2006.




Lin, C. C., Chen, S. J., Huang, K. L., Hwang, W. I., Chang-Chien, G. P., and Lin, W. Y.:
Characteristics of metals in nano/ultrafine/fine/coarse particles collected beside a heavily
trafficked road, Environ Sci Technol, 39, 8113-8122, 10.1021/es048182a, 2005.
Linak, W. P., and Miller, C. A.: Comparison of particle size distributions and elemental
partitioning from the combustion of pulverized coal and residual fuel oil, J Air Waste Manage, 50,
1532-1544, Doi 10.1080/10473289.2000.10464171, 2000.
Ma, Y., Li, S., Zheng, J., Khalizov, A., Wang, X., Wang, Z., and Zhou, Y.: Size-resolved
measurements of mixing state and cloud-nucleating ability of aerosols in Nanjing, China, Journal
of Geophysical Research: Atmospheres, 122, 9430-9450, 10.1002/2017jd026583, 2017.
Mahowald, N., Jickells, T. D., Baker, A. R., Artaxo, P., Benitez-Nelson, C. R., Bergametti, G.,
Bond, T. C., Chen, Y., Cohen, D. D., Herut, B., Kubilay, N., Losno, R., Luo, C., Maenhaut, W.,
McGee, K. A., Okin, G. S., Siefert, R. L., and Tsukuda, S.: Global distribution of atmospheric
phosphorus sources, concentrations and deposition rates, and anthropogenic impacts, Global
Biogeochem Cy, 22, 10.1029/2008gb003240, 2008.
Marple, V., Olson, B., Romay, F., Hudak, G., Geerts, S. M. and Lundgren, D.: Second generation
micro-orifice uniform deposit impactor, 120 MOUDI-II: Design, Evaluation, and application to
long-term ambient sampling, Aerosol Sci. Technol., 48(4), 427–433,
doi:10.1080/02786826.2014.884274, 2014.
Martens, C. S., Wesolowski, J. J., Harriss, R. C., and Kaifer, R.: Chlorine Loss from Puerto-Rican
and San-Francisco-Bay Area Marine Aerosols, J Geophys Res, 78, 8778-8792, DOI
10.1029/JC078i036p08778, 1973.
Maudlin, L. C., Wang, Z., Jonsson, H. H., and Sorooshian, A.: Impact of wildfires on size-
resolved aerosol composition at a coastal California site, Atmos Environ, 119, 59-68,
10.1016/j.atmosenv.2015.08.039, 2015.
Metcalf, A. R., Craven, J. S., Ensberg, J. J., Brioude, J., Angevine, W., Sorooshian, A., Duong,
H. T., Jonsson, H. H., Flagan, R. C., and Seinfeld, J. H.: Black carbon aerosol over the Los
Angeles Basin during CalNex, J Geophys Res-Atmos, 117, 10.1029/2011jd017255, 2012.
Mielonen, T., Levy, R. C., Aaltonen, V., Komppula, M., de Leeuw, G., Huttunen, J., Lihavainen,
H., Kolmonen, P., Lehtinen, K. E. J., and Arola, A.: Evaluating the assumptions of surface
reflectance and aerosol type selection within the MODIS aerosol retrieval over land: the problem
of dust type selection, Atmos Meas Tech, 4, 201-214, 10.5194/amt-4-201-2011, 2011.
Miller, J., and Miller, J.C.: Statistics and chemometrics for analytical chemistry. Pearson
Education, 2018.
Mishra, S. K., Agnihotri, R., Yadav, P. K., Singh, S., Prasad, M. V. S. N., Praveen, P. S., Tawale,
J. S., Rashmi, Mishra, N. D., Arya, B. C., and Sharma, C.: Morphology of Atmospheric Particles
over Semi-Arid Region (Jaipur, Rajasthan) of India: Implications for Optical Properties, Aerosol
Air Qual Res, 15, 974-+, 10.4209/aaqr.2014.10.0244, 2015.






Mooibroek, D., Schaap, M., Weijers, E. P., and Hoogerbrugge, R.: Source apportionment and
spatial variability of PM$_{2.5}$ using measurements at five sites in the Netherlands, Atmospheric
Environment, 45, 4180-4191, 10.1016/j.atmosenv.2011.05.017, 2011.

Mosier, A. R., Andre, C. E., and Viets, F. G.: Identification of Aliphatic-Amines Volatilized from
Cattle Feedyard, Environ Sci Technol, 7, 642-644, DOI 10.1021/es60079a009, 1973.

Muller, C., Iinuma, Y., Karstensen, J., van Pinxteren, D., Lehmann, S., Gnauk, T., and Herrmann,
H.: Seasonal variation of aliphatic amines in marine sub-micrometer particles at the Cape Verde
islands, Atmos Chem Phys, 9, 9587-9597, 2009.

Murphy, S. M., Agrawal, H., Sorooshian, A., Padro, L. T., Gates, H., Hersey, S., Welch, W. A.,
Jung, H., Miller, J. W., Cocker, D. R., Nenes, A., Jonsson, H. H., Flagan, R. C., and Seinfeld, J.
H.: Comprehensive Simultaneous Shipboard and Airborne Characterization of Exhaust from a
Modern Container Ship at Sea, Environ Sci Technol, 43, 4626-4640, 10.1021/es802413j, 2009.

Norris, G., Duvall, R., Brown, S., and Bai, S.: EPA Positive Matrix Factorization (PMF) 5.0
fundamentals and User Guide Prepared for the US Environmental Protection Agency Office of
Research and Development, Washington, DC. Inc., Petaluma, 2014.

Nriagu, J. O.: A Global Assessment of Natural Sources of Atmospheric Trace-Metals, Nature, 338,
47-49, DOI 10.1038/338047a0, 1989.

Pabroa, P. C. B., Santos, F. L., Morco, R. P., Racho, J. M. D., Bautista, A. T., and Bucal, C. G. D.:
Receptor modeling studies for the characterization of air particulate lead pollution sources in
Valenzuela sampling site (Philippines), Atmos Pollut Res, 2, 213-218, 10.5094/Apr.2011.027,
1005  2011.


Philippine Statistics Authority: https://psa.gov.ph/, Accessed 28 August 2018.

Polissar, A., Hopke, P., Paatero, P., Malm, W., and Sisler, J.: Atmospheric aerosol over Alaska 2.
Elemental composition and sources. Journal of Geophysical Research 103, 19045-19057, 1998.

Prabhakar, G., Ervens, B., Wang, Z., Maudlin, L. C., Coggon, M. M., Jonsson, H. H., Seinfeld, J.
H., and Sorooshian, A.: Sources of nitrate in stratocumulus cloud water: Airborne measurements
during the 2011 E-PEACE and 2013 NiCE studies, Atmos Environ, 97, 166-173,
10.1016/j.atmosenv.2014.08.019, 2014a.

Prabhakar, G., Sorooshian, A., Toffol, E., Arellano, A. F., and Betterton, E. A.: Spatiotemporal
distribution of airborne particulate metals and metalloids in a populated arid region, Atmos
Environ, 92, 339-347, 10.1016/j.atmosenv.2014.04.044, 2014b.

Qu, W. J., Wang, J., Zhang, X. Y., Wang, D., and Sheng, L. F.: Influence of relative humidity on
aerosol composition: Impacts on light extinction and visibility impairment at two sites in coastal
area of China, Atmos Res, 153, 500-511, 10.1016/j.atmosres.2014.10.009, 2015.



Raatikainen, T., Brus, D., Hyvärinen, A. P., Svensson, J., Asmi, E., and Lihavainen, H.: Black
carbon concentrations and mixing state in the Finnish Arctic, Atmos. Chem. Phys., 15, 10057-
10070, 10.5194/acp-15-10057-2015, 2015.
Ramachandran, S., and Rajesh, T. A.: Black carbon aerosol mass concentrations over
Ahmedabad, an urban location in western India: Comparison with urban sites in Asia, Europe,
Canada, and the United States, J Geophys Res-Atmos, 112, 10.1029/2006jd007488, 2007.
Ran, L., Deng, Z. Z., Wang, P. C., and Xia, X. A.: Black carbon and wavelength-dependent aerosol
absorption in the North China Plain based on two-year aethalometer measurements, Atmos
Environ, 142, 132-144, 10.1016/j.atmosenv.2016.07.014, 2016.
Reddington, C. L., McMeeking, G., Mann, G. W., Coe, H., Frontoso, M. G., Liu, D., Flynn, M.,
Spracklen, D. V., and Carslaw, K. S.: The mass and number size distributions of black carbon
aerosol over Europe, Atmos. Chem. Phys., 13, 4917-4939, 10.5194/acp-13-4917-2013, 2013.
Reff, A., Eberly, S.I., and Bhave, P.V.: Receptor modeling of ambient particulate matter data using
positive matrix factorization: Review of existing methods. J Air Waste Manage 57, 146-154, 2007.
Reid, J. S., Xian, P., Hyer, E. J., Flatau, M. K., Ramirez, E. M., Turk, F. J., Sampson, C. R., Zhang,
C., Fukada, E. M., and Maloney, E. D.: Multi-scale meteorological conceptual analysis of observed
active fire hotspot activity and smoke optical depth in the Maritime Continent, Atmos Chem Phys,
12, 2117-2147, 10.5194/acp-12-2117-2012, 2012.
Reid, J. S., Hyer, E. J., Johnson, R. S., Holben, B. N., Yokelson, R. J., Zhang, J. L., Campbell, J.
R., Christopher, S. A., Di Girolamo, L., Giglio, L., Holz, R. E., Kearney, C., Miettinen, J., Reid,
E. A., Turk, F. J., Wang, J., Xian, P., Zhao, G. Y., Balasubramanian, R., Chew, B. N., Janjai, S.,
Lagrosas, N., Lestari, P., Lin, N. H., Mahmud, M., Nguyen, A. X., Norris, B., Oanh, N. T. K., Oo,
M., Salinas, S. V., Welton, E. J., and Liew, S. C.: Observing and understanding the Southeast
Asian aerosol system by remote sensing: An initial review and analysis for the Seven Southeast
Asian Studies (7SEAS) program, Atmos Res, 122, 403-468, 10.1016/j.atmosres.2012.06.005,
1056  2013.

Reid, J. S., Xian, P., Holben, B. N., Hyer, E. J., Reid, E. A., Salinas, S. V., Zhang, J. L., Campbell,
J. R., Chew, B. N., Holz, R. E., Kuciauskas, A. P., Lagrosas, N., Posselt, D. J., Sampson, C. R.,
Walker, A. L., Welton, E. J., and Zhang, C. D.: Aerosol meteorology of the Maritime Continent
for the 2012 7SEAS southwest monsoon intensive study - Part 1: regional-scale phenomena,
Atmos Chem Phys, 16, 14041-14056, 10.5194/acp-16-14041-2016, 2016a.
Reid, J. S., Lagrosas, N. D., Jonsson, H. H., Reid, E. A., Atwood, S. A., Boyd, T. J., Ghate, V. P.,
Xian, P., Posselt, D. J., Simpas, J. B., Uy, S. N., Zaiger, K., Blake, D. R., Bucholtz, A., Campbell,
J. R., Chew, B. N., Cliff, S. S., Holben, B. N., Holz, R. E., Hyer, E. J., Kreidenweis, S. M.,
Kuciauskas, A. P., Lolli, S., Oo, M., Perry, K. D., Salinas, S. V., Sessions, W. R., Smirnov, A.,
Walker, A. L., Wang, Q., Yu, L. Y., Zhang, J. L., and Zhao, Y. J.: Aerosol meteorology of
Maritime Continent for the 2012 7SEAS southwest monsoon intensive study - Part 2: Philippine



receptor observations of fine-scale aerosol behavior, Atmos Chem Phys, 16, 14057-14078, 10.5194/acp-16-14057-2016, 2016b.

Rocha, L. D. S., and Correa, S. M.: Determination of size-segregated elements in diesel-biodiesel blend exhaust emissions, Environ Sci Pollut R, 25, 18121-18129, 10.1007/s11356-018-1980-8, 2018.

Rogge, W. F., Mazurek, M. A., Hildemann, L. M., Cass, G. R., and Simoneit, B. R. T.: Quantification of Urban Organic Aerosols at a Molecular-Level - Identification, Abundance and Seasonal-Variation, Atmos Environ a-Gen, 27, 1309-1330, Doi 10.1016/0960-1686(93)90257-Y, 1993.

Ro, C. U., Oh, K. Y., Kim, H., Kim, Y. P., Lee, C. B., Kim, K. H., Kang, C. H., Osan, J., De Hoog, J., Worobiec, A., and Van Grieken, R.: Single-particle analysis of aerosols at Cheju Island, Korea, using low-Z electron probe X-ray microanalysis: A direct proof of nitrate formation from sea salts, Environ Sci Technol, 35, 4487-4494, 10.1021/es0155231, 2001.

Rolph, G.D.: Real-time Environmental Applications and Display sYstem (READY) website (http://ready.Arl.NOAA.Gov), NOAA Air Resour. Lab., Silver Spring, Md., 2016.

Roth, B., and Okada, K.: On the modification of sea-salt particles in the coastal atmosphere, Atmos Environ, 32, 1555-1569, Doi 10.1016/S1352-2310(97)00378-6, 1998.

Saltzman, E. S., Savoie, D. L., Zika, R. G., and Prospero, J. M.: Methane Sulfonic-Acid in the Marine Atmosphere, J Geophys Res-Oceans, 88, 897-902, DOI 10.1029/JC088iC15p10897, 1983.

Saltzman, E. S., Savoie, D. L., Prospero, J. M., and Zika, R. G.: Methanesulfonic-Acid and Non-Sea-Salt Sulfate in Pacific Air - Regional and Seasonal-Variations, J Atmos Chem, 4, 227-240, Doi 10.1007/Bf00052002, 1986.

Schade, G. W., and Crutzen, P. J.: Emission of Aliphatic-Amines from Animal Husbandry and Their Reactions - Potential Source of N$_2$O and HCN, J Atmos Chem, 22, 319-346, Doi 10.1007/Bf00696641, 1995.

Schwarz, J. P., Gao, R. S., Spackman, J. R., Watts, L. A., Thomson, D. S., Fahey, D. W., Ryerson, T. B., Peischl, J., Holloway, J. S., Trainer, M., Frost, G. J., Baynard, T., Lack, D. A., de Gouw, J. A., Warneke, C., and Del Negro, L. A.: Measurement of the mixing state, mass, and optical size of individual black carbon particles in urban and biomass burning emissions, Geophysical Research Letters, 35, 10.1029/2008gl033968, 2008.

Seinfeld, J. H., and Pandis, S. N.: Atmospheric chemistry and physics (3rd ed.). New York: Wiley-Interscience, 2016.

Shafer, M. M., Toner, B. M., Oyerdier, J. T., Schauer, J. J., Fakra, S. C., Hu, S. H., Herner, J. D., and Ayala, A.: Chemical Speciation of Vanadium in Particulate Matter Emitted from Diesel





Vehicles and Urban Atmospheric Aerosols, Environ Sci Technol, 46, 189-195,
10.1021/es200463c, 2012.
Shingler, T., Sorooshian, A., Ortega, A., Crosbie, E., Wonaschutz, A., Perring, A. E.,
Beyersdorf, A., Ziemba, L., Jimenez, J. L., Campuzano-Jost, P., Mikoviny, T., Wisthaler, A., and
Russell, L. M.: Ambient observations of hygroscopic growth factor and f(RH) below 1: Case
studies from surface and airborne measurements, J Geophys Res-Atmos, 121, 13661-13677,
10.1002/2016jd025471, 2016.
Shiraiwa, M., Kondo, Y., Moteki, N., Takegawa, N., Sahu, L. K., Takami, A., Hatakeyama, S.,
Yonemura, S., and Blake, D. R.: Radiative impact of mixing state of black carbon aerosol in Asian
outflow, Journal of Geophysical Research: Atmospheres, 113, 10.1029/2008jd010546, 2008.
Simpas, J., Lorenzo, G., and Cruz, M. T.: Monitoring Particulate Matter Levels and Composition
for Source Apportionment Study in Metro Manila, Philippines, in: Improving Air Quality in Asian
Developing Countries: Compilation of Research Findings, Kim Oanh, N. T. (Ed.), NARENCA,
Vietnam Publishing House of Natural Resources, Environment and Cartography, Vietnam, 239-
1132    261, 2014.
Singh, M., Jaques, P. A., and Sioutas, C.: Size distribution and diurnal characteristics of particle-
bound metals in source and receptor sites of the Los Angeles Basin, Atmos Environ, 36, 1675-
1689, Pii S1352-2310(02)00166-8, Doi 10.1016/S1352-2310(02)00166-8, 2002.
Song, F., and Gao, Y.: Size distributions of trace elements associated with ambient particular
matter in the affinity of a major highway in the New Jersey-New York metropolitan area, Atmos
Environ, 45, 6714-6723, 10.1016/j.atmosenv.2011.08.031, 2011.
Sorooshian, A., Ng, N. L., Chan, A. W. H., Feingold, G., Flagan, R. C., and Seinfeld, J. H.:
Particulate organic acids and overall water-soluble aerosol composition measurements from the
2006 Gulf of Mexico Atmospheric Composition and Climate Study (GoMACCS), J Geophys
Res-Atmos, 112, 10.1029/2007jd008537, 2007.
Sorooshian, A., Murphy, S. N., Hersey, S., Gates, H., Padro, L. T., Nenes, A., Brechtel, F. J.,
Jonsson, H., Flagan, R. C., and Seinfeld, J. H.: Comprehensive airborne characterization of aerosol
from a major bovine source, Atmos Chem Phys, 8, 5489-5520, DOI 10.5194/acp-8-5489-2008,
1150    2008.
Sorooshian, A., Padro, L. T., Nenes, A., Feingold, G., McComiskey, A., Hersey, S. P., Gates, H.,
Jonsson, H. H., Miller, S. D., Stephens, G. L., Flagan, R. C., and Seinfeld, J. H.: On the link
between ocean biota emissions, aerosol, and maritime clouds: Airborne, ground, and satellite
measurements off the coast of California, Global Biogeochem Cy, 23,
10.1029/2009gb003464, 2009.
Sorooshian, A., Crosbie, E., Maudlin, L. C., Youn, J. S., Wang, Z., Shingler, T., Ortega, A. M.,
Hersey, S., and Woods, R. K.: Surface and airborne measurements of organosulfur and
methanesulfonate over the western United States and coastal areas, J Geophys Res-Atmos, 120,
8535-8548, 10.1002/2015jd023822, 2015.



Stein, A. F., Draxler, R. R., Rolph, G. D., Stunder, B. J. B., Cohen, M. D., and Ngan, F.: NOAA's Hysplit Atmospheric Transport and Dispersion Modeling System, B Am Meteorol Soc, 96, 2059-2077, 10.1175/Bams-D-14-00110.1, 2015.

Tai, A. P. K., Mickley, L. J., and Jacob, D. J.: Correlations between fine particulate matter (PM2.5) and meteorological variables in the United States: Implications for the sensitivity of PM2.5 to climate change, Atmos Environ, 44, 3976-3984, 10.1016/j.atmosenv.2010.06.060, 2010.

Tsakalou, C., Papamarkou, S., Tsakiridis, P. E., Bartzas, G., and Tsakalakis, K.: Characterization and leachability evaluation of medical wastes incineration fly and bottom ashes and their vitrification outgrowths, J Environ Chem Eng, 6, 367-376, 10.1016/j.jece.2017.12.012, 2018.

VandenBoer, T. C., Petroff, A., Markovic, M. Z., and Murphy, J. G.: Size distribution of alkyl amines in continental particulate matter and their online detection in the gas and particle phase, Atmos Chem Phys, 11, 4319-4332, 10.5194/acp-11-4319-2011, 2011.

Villafuerte, M. Q., Matsumoto, J., Akasaka, I., Takahashi, H. G., Kubota, H., and Cinco, T. A.: Long-term trends and variability of rainfall extremes in the Philippines, Atmos Res, 137, 1-13, 10.1016/j.atmosres.2013.09.021, 2014.

Vossler, T., Cernikovsky, L., Novak, J., and Williams, R.: Source apportionment with uncertainty estimates of fine particulate matter in Ostrava, Czech Republic using Positive Matrix Factorization, Atmos Pollut Res, 7, 503-512, 10.1016/j.apr.2015.12.004, 2016.

Wang, J., Ge, C., Yang, Z. F., Hyer, E. J., Reid, J. S., Chew, B. N., Mahmud, M., Zhang, Y. X., and Zhang, M. G.: Mesoscale modeling of smoke transport over the Southeast Asian Maritime Continent: Interplay of sea breeze, trade wind, typhoon, and topography, Atmos Res, 122, 486-503, 10.1016/j.atmosres.2012.05.009, 2013.

Wang, Y. Q., Zhang, X. Y., and Draxler, R. R.: TrajStat: GIS-based software that uses various trajectory statistical analysis methods to identify potential sources from long-term air pollution measurement data, Environ Modell Softw, 24, 938-939, 10.1016/j.envsoft.2009.01.004, 2009.

Wasson, S. J., Linak, W. P., Gullett, B. K., King, C. J., Touati, A., Huggins, F. E., Chen, Y. Z., Shah, N., and Huffman, G. P.: Emissions of chromium, copper, arsenic, and PCDDs/Fs from open burning of CCA-treated wood, Environ Sci Technol, 39, 8865-8876, 10.1021/es050891g, 2005.

Watson, J. G.: Protocol for Applying and Validating the CMB Model for PM$_{2.5}$ and VOC. Report No. EPA-451/R-04-001. US Environmental Protection Agency, Research Triangle Park, NC., 2004.

Watts, S. F., Watson, A., and Brimblecombe, P.: Measurements of the Aerosol Concentrations of Methanesulfonic Acid, Dimethyl-Sulfoxide and Dimethyl Sulfone in the Marine Atmosphere of the British-Isles, Atmos Environ, 21, 2667-2672, Doi 10.1016/0004-6981(87)90198-3, 1987.





Wonaschuetz, A., Sorooshian, A., Ervens, B., Chuang, P. Y., Feingold, G., Murphy, S. M., de
Gouw, J., Warneke, C., and Jonsson, H. H.: Aerosol and gas re-distribution by shallow cumulus
clouds: An investigation using airborne measurements, J Geophys Res-Atmos, 117,
10.1029/2012jd018089, 2012.
Wu, D., Zhang, F., Lou, W. H., Li, D., and Chen, J. M.: Chemical characterization and toxicity
assessment of fine particulate matters emitted from the combustion of petrol and diesel fuels, Sci
Total Environ, 605, 172-179, 10.1016/j.scitotenv.2017.06.058, 2017.
Xian, P., Reid, J. S., Atwood, S. A., Johnson, R. S., Hyer, E. J., Westphal, D. L., and Sessions, W.:
Smoke aerosol transport patterns over the Maritime Continent, Atmos Res, 122, 469-485,
10.1016/j.atmosres.2012.05.006, 2013.
Xu, G. J., and Gao, Y.: Characterization of marine aerosols and precipitation through shipboard
observations on the transect between 31 degrees N-32 degrees S in the West Pacific, Atmos Pollut
Res, 6, 154-161, 10.5094/Apr.2015.018, 2015.
Yao, X. H., Fang, M., and Chan, C. K.: The size dependence of chloride depletion in fine and
coarse sea-salt particles, Atmos Environ, 37, 743-751, 10.1016/S1352-2310(02)00955-X, 2003.
Youn, J. S., Wang, Z., Wonaschutz, A., Arellano, A., Betterton, E. A., and Sorooshian, A.:
Evidence of aqueous secondary organic aerosol formation from biogenic emissions in the North
American Sonoran Desert, Geophys Res Lett, 40, 3468-3472, 10.1002/grl.50644, 2013.
Youn, J. S., Crosbie, E., Maudlin, L. C., Wang, Z., and Sorooshian, A.: Dimethylamine as a major
alkyl amine species in particles and cloud water: Observations in semi-arid and coastal regions,
Atmos Environ, 122, 250-258, 10.1016/j.atmosenv.2015.09.061, 2015.
Youn, J. S., Csavina, J., Rine, K. P., Shingler, T., Taylor, M. P., Saez, A. E., Betterton, E. A., and
Sorooshian, A.: Hygroscopic Properties and Respiratory System Deposition Behavior of
Particulate Matter Emitted By Mining and Smelting Operations, Environ Sci Technol, 50, 11706-
11713, 10.1021/acs.est.6b03621, 2016.





**Table 1.** Summary of average operating parameters, meteorological conditions, and total resolved water-soluble mass concentration for each MOUDI sample set collected at Manila Observatory (MO) during the 2018 Southwest Monsoon period. On two occasions, simultaneous MOUDI sets were collected for one set to undergo gravimetric analysis (MO3 and MO13) to compare with mass resolved from chemical speciation of the water-soluble fraction (MO4 and MO14). One additional MOUDI set devoted to microscopy analysis was collected using aluminum substrates for one hour on August 1 at 30 LPM.

| Sample set name | Dates | Duration (hrs) | Flow rate (LPM) | Wind speed (m/s) | Wind direction (°) | T (°C) | Rain (mm) | Water-soluble mass ($\mu g\ m^{-3}$) |
|---|---|---|---|---|---|---|---|---|
| MO1 | Jul 19-20 | 24 | 30 | 3.3 | 90.1 | 24.9 | 47 | 4.6 |
| MO2 | Jul 23-25 | 54 | 30 | 1.3 | 95.8 | 26.7 | 7.8 | 6.5 |
| MO3/4 | Jul 25-30 | 119 | 28/30 | 1.2 | 111.8 | 26.7 | 49.6 | 5.2 |
| MO5 | Jul 30-Aug 1 | 42 | 29 | 2.6 | 98.1 | 27.5 | 52.8 | 9.2 |
| MO6 | Aug 6-8 | 48 | 27 | 0.9 | 127.5 | 26.1 | 30.4 | 5.1 |
| MO7 | Aug 14-16 | 48 | 28 | 3.0 | 107.8 | 27.8 | 2.8 | 13.7 |
| MO8 | Aug 22-24 | 48 | 29 | 3.5 | 108.7 | 28.1 | 1 | 12.8 |
| MO9 | Sep 1-3 | 48 | 27 | 0.7 | 98.6 | 26.6 | 51.6 | 6.2 |
| MO10 | Sep 10–12 | 48 | 29 | 1.0 | 94.7 | 26.2 | 78.4 | 6.4 |
| MO11 | Sep 18–20 | 48 | 27 | 0.5 | 290.2 | 27.8 | 0 | 2.7 |
| MO12 | Sep 26-28 | 48 | 27 | 1.2 | 96.3 | 27.8 | 6.8 | 13.5 |
| MO13/14 | Oct 6-8 | 48 | 28/26 | 0.6 | 108.2 | 27.8 | 0.8 | 16.6 |



**Table 2**. Charge balance slopes (cations on y-axis; anions on x-axis) for the MOUDI sets shown
including the averages of all sets (All) for three size ranges: submicrometer stages spanning 0.056
– 1.0 µm; supermicrometer stages (> 1.0 µm); and all stages (> 0.056 µm). The species used in
the charge balance analysis include those speciated with the IC (listed in Section 2.3) plus K from
ICP-QQQ analysis.

| Sample set | 0.056 – 1.0 µm | > 1 µm | > 0.056 µm |
|---|---|---|---|
| MO1 | 0.87 | 1.37 | 0.89 |
| MO2 | 1.46 | 1.26 | 1.41 |
| MO4 | 1.25 | 1.17 | 1.21 |
| MO5 | 1.35 | 1.43 | 1.41 |
| MO6 | 1.29 | 1.45 | 1.31 |
| MO7 | 1.40 | 1.23 | 1.36 |
| MO8 | 1.35 | 1.33 | 1.36 |
| MO9 | 1.28 | 1.55 | 1.26 |
| MO10 | 1.37 | 1.36 | 1.35 |
| MO11 | 0.97 | 1.60 | 1.27 |
| MO12 | 1.37 | 1.19 | 1.33 |
| MO14 | 1.31 | 1.28 | 1.29 |
| All | 1.35 | 1.24 | 1.33 |






**Table 3.** Contributions (in weight percentage) of each PMF source factor to the total mass in
different diameter ranges.

| Diameter Range (μm) | Aged/ Transported | Sea Salt | Combustion | Vehicular/ Resuspended Dust | Waste Processing |
|---|---|---|---|---|---|
| > 0.056 | 48.0% | 22.5% | 18.7% | 5.6% | 5.1% |
| 0.056 - 1.0 | 68.9% | 0.6% | 23.9% | 1.5% | 5.1% |
| > 1.0 | 18.6% | 53.5% | 11.3% | 11.3% | 5.3% |




**Table 4.** Correlation matrix (r values) between water-soluble species based on total MOUDI-
integrated mass concentrations (> 0.056 μm). Blank cells represent statistically insignificant
values. Results for the sub- and supermicrometer ranges are in Tables S2-S3. Panels A-E
represent important species from each of the source profiles identified in Section 3.3: A =
Aged/Transported, B = Sea Salt, C = Combustion, D = Vehicular/Resuspended Dust, E = Waste
Processing. DMA – Dimethylamine, MSA – Methanesulfonate, PH – Phthalate, OX – Oxalate,
MA – Maleate, SU – Succinate, AD – Adipate.

**A)**

|  | OX | SO₄ | NH₄ | Sn | Rb | K | Cs | V | DMA | MSA | PH | SU | AD | Se | Tl |
|---|---|---|---|---|---|---|---|---|---|---|---|---|---|---|---|
| OX | 1.00 | | | | | | | | | | | | | | |
| SO₄ | 0.74 | 1.00 | | | | | | | | | | | | | |
| NH₄ | 0.68 | 0.99 | 1.00 | | | | | | | | | | | | |
| Sn | 0.71 | 0.87 | 0.85 | 1.00 | | | | | | | | | | | |
| Rb | 0.73 | 0.74 | 0.73 | 0.69 | 1.00 | | | | | | | | | | |
| K | 0.76 | 0.71 | 0.69 | 0.69 | 0.97 | 1.00 | | | | | | | | | |
| Cs | 0.72 | 0.82 | 0.81 | 0.74 | 0.96 | 0.91 | 1.00 | | | | | | | | |
| V | 0.36 | 0.64 | 0.63 | 0.48 | 0.53 | 0.51 | 0.57 | 1.00 | | | | | | | |
| DMA | | 0.35 | | 0.38 | 0.45 | 0.37 | 0.45 | | 1.00 | | | | | | |
| MSA | 0.71 | 0.89 | 0.89 | 0.79 | 0.90 | 0.85 | 0.92 | 0.51 | 0.47 | 1.00 | | | | | |
| PH | 0.68 | 0.67 | 0.68 | 0.73 | 0.82 | 0.76 | 0.80 | | 0.38 | 0.88 | 1.00 | | | | |
| SU | 0.63 | 0.56 | 0.59 | 0.44 | 0.87 | 0.81 | 0.82 | | 0.68 | 0.78 | 0.84 | 1.00 | | | |
| AD | 0.40 | 0.66 | 0.70 | 0.62 | 0.70 | 0.70 | 0.77 | | 0.84 | 0.74 | 0.75 | 0.90 | 1.00 | | |
| Se | 0.75 | 0.75 | 0.73 | 0.66 | 0.80 | 0.78 | 0.79 | 0.32 | 0.34 | 0.78 | 0.80 | 0.88 | 0.88 | 1.00 | |
| Tl | 0.75 | 0.87 | 0.86 | 0.80 | 0.89 | 0.85 | 0.94 | 0.74 | 0.65 | 0.80 | 0.52 | 0.70 | | 0.43 | 1.00 |

**B)**

|  | Cl | NO₃ | Ba | Sr | Ca | Na | Mg | Hf |
|---|---|---|---|---|---|---|---|---|
| Cl | 1.00 | | | | | | | |
| NO₃ | 0.76 | 1.00 | | | | | | |
| Ba | 0.66 | 0.80 | 1.00 | | | | | |
| Sr | 0.78 | 0.87 | 0.91 | 1.00 | | | | |
| Ca | 0.58 | 0.79 | 0.75 | 0.78 | 1.00 | | | |
| Na | 0.93 | 0.87 | 0.75 | 0.85 | 0.63 | 1.00 | | |
| Mg | 0.91 | 0.87 | 0.77 | 0.87 | 0.66 | 0.99 | 1.00 | |
| Hf | | | | | 0.57 | | | 1.00 |

**C)**

|  | As | Ni | Co | P | Mo | Cr | Mal | Ag |
|---|---|---|---|---|---|---|---|---|
| As | 1.00 | | | | | | | |
| Ni | 0.58 | 1.00 | | | | | | |
| Co | | | 1.00 | | | | | |
| P | | 0.33 | 0.34 | 1.00 | | | | |
| Mo | | | | | 1.00 | | | |
| Cr | 0.62 | 0.49 | | 0.20 | | 1.00 | | |
| MA | | | 0.67 | | -0.42 | | 1.00 | |
| Ag | | | 0.85 | | 0.64 | | | 1.00 |

**D)**

|  | Zr | Y | Al | Fe | Ti | Nb |
|---|---|---|---|---|---|---|
| Zr | 1.00 | | | | | |
| Y | 0.75 | 1.00 | | | | |
| Al | 0.88 | 0.76 | 1.00 | | | |
| Fe | 0.33 | 0.61 | 0.25 | 1.00 | | |
| Ti | 0.84 | 0.66 | 0.82 | 0.41 | 1.00 | |
| Nb | 0.70 | 0.50 | 0.59 | 0.59 | 0.70 | 1.00 |

**E)**

|  | Cd | Zn | Cu | Mn | Pb |
|---|---|---|---|---|---|
| Cd | 1.00 | | | | |
| Zn | 0.60 | 1.00 | | | |
| Cu | 0.21 | 0.27 | 1.00 | | |
| Mn | 0.28 | 0.61 | 0.22 | 1.00 | |
| Pb | 0.78 | 0.58 | 0.38 | 0.27 | 1.00 |


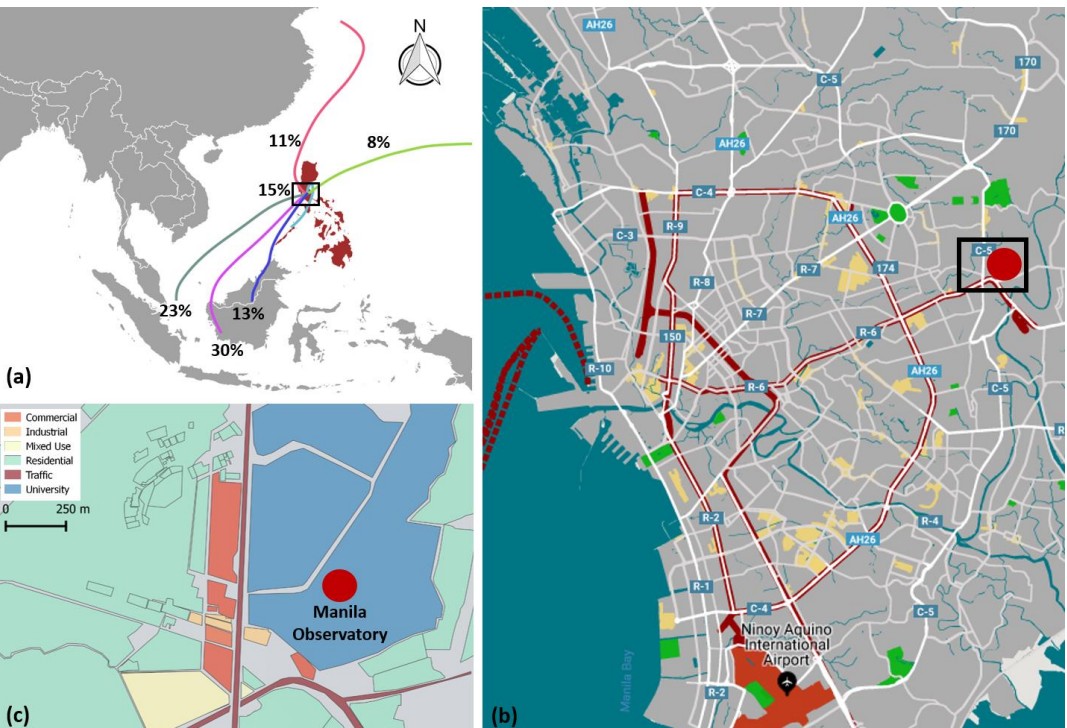

**Figure 1.** (a) Location of Metro Manila, Philippines relative to Southeast Asia. Also shown are 5-day backward trajectory frequencies during the sampling duration based on HYSPLIT cluster analysis; note that 15% correspond to trajectories within the black square. (b) Close-up view of Metro Manila showing the location of the Manila Observatory sampling site with a black rectangle. The base map shows roads, commercial centers, and major transit lines in the city. (c) Land use classification in the vicinity of the sampling site. (Sources: GADM, Snazzy Maps, OpenStreetMap, NOAA HYSPLIT, & TrajSat)



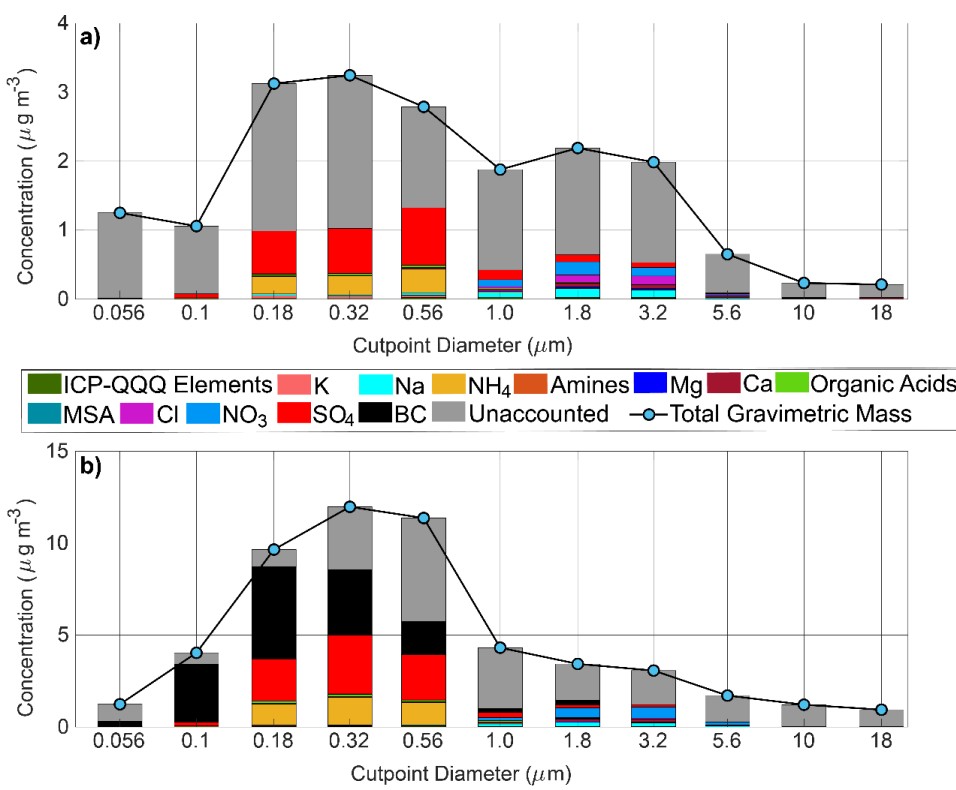

**Figure 2.** Mass size distributions of total PM (blue markers) and resolved chemical species
(colored bars) for MOUDI sets (a) MO3/4 and (b) MO13/14. Note that set MO13 was the single
MOUDI set where BC was quantified. ICP-QQQ = sum of water-soluble elements except K;
amines = sum of DMA, TMA, DEA; organic acids = sum of oxalate, succinate, adipate,
pyruvate, phthalate, maleate.




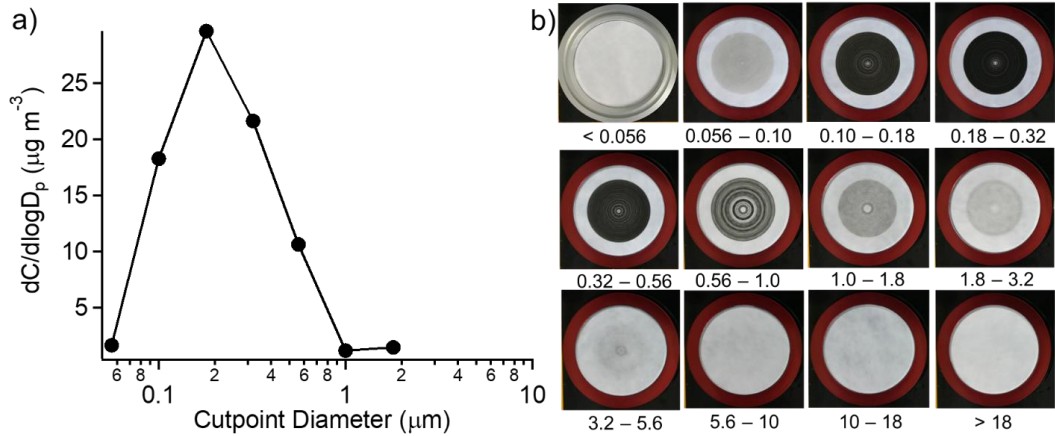

**Figure 3.** (a) Mass size distribution of BC retrieved from the MABI optical measurement at 870
nm for set MO13. Missing values were below detection limits. (b) Photographs of each stage of
set MO13 with numbers below each image representing the aerodynamic diameter ranges in units
of μm.





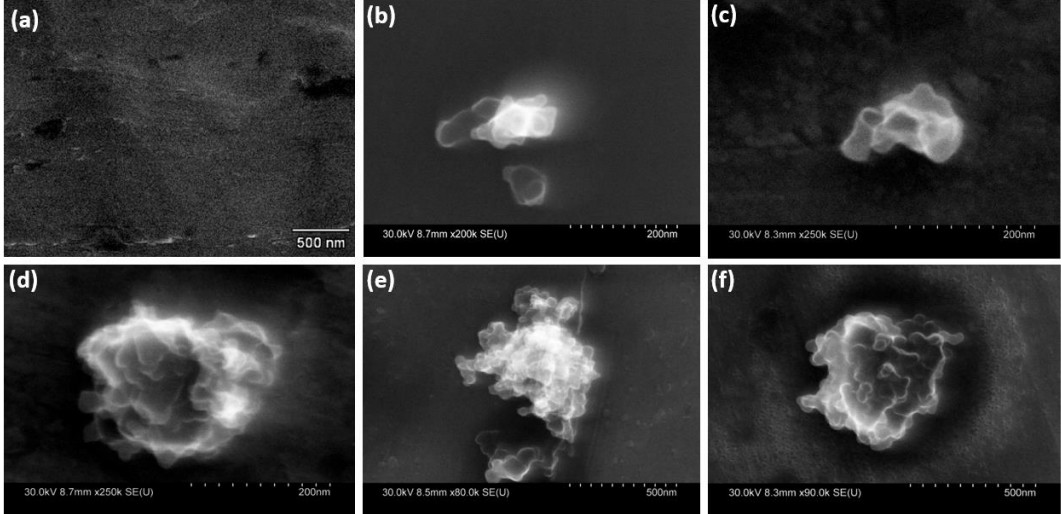

**Figure 4.** SEM image of a (a) blank filter and (b-f) individual particles in different sub-micrometer aerodynamic diameter ranges sampled by the MOUDI: (b) 0.056–0.1 μm, (c) 0.1–0.18 μm, (d) 0.18–0.32 μm, (e) 0.32–0.56 μm, (f) 0.56–1.0 μm.





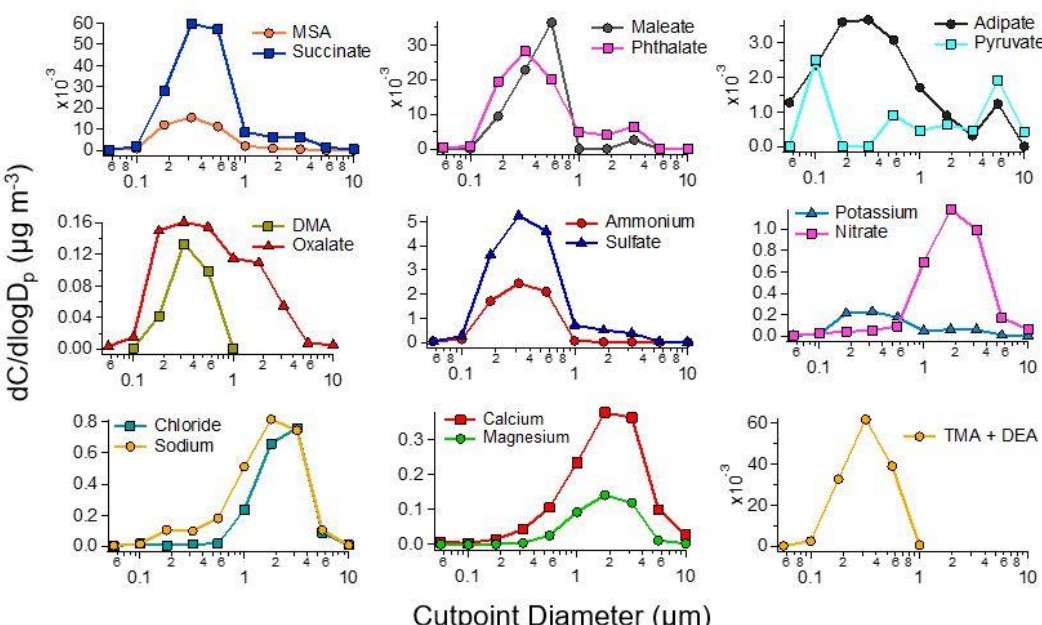

**Figure 5.** Average mass size distribution of water-soluble ions speciated via IC in addition to
potassium from ICP-QQQ analysis.




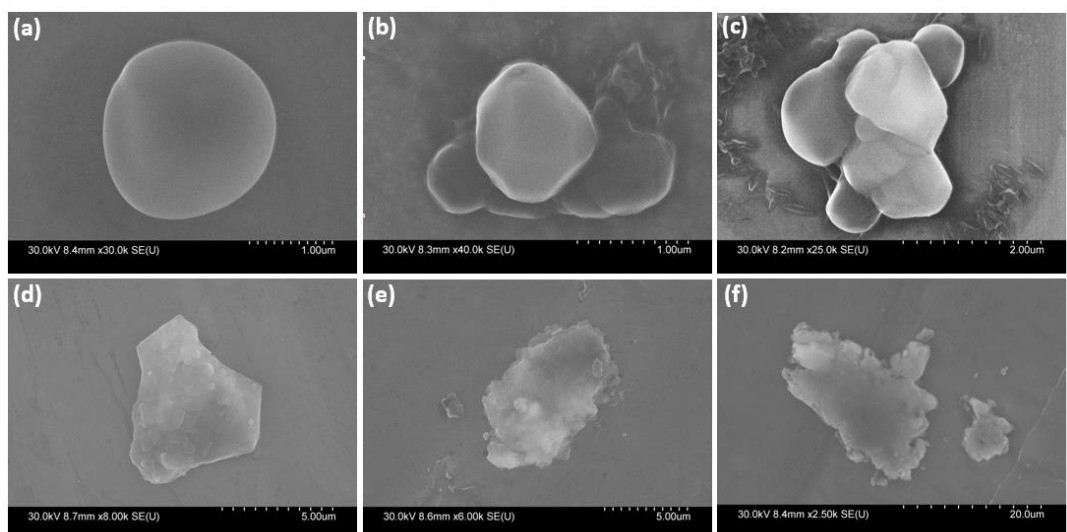

**Figure 6.** Same as Figure 4, but for different supermicrometer aerodynamic diameter ranges
sampled by the MOUDI: (a) 1.0–1.8 µm, (b) 1.8–3.2 µm; (c) 3.2–5.6 µm, (d) 5.6–10 µm, (e) 10-
18 µm, (f) > 18 µm.



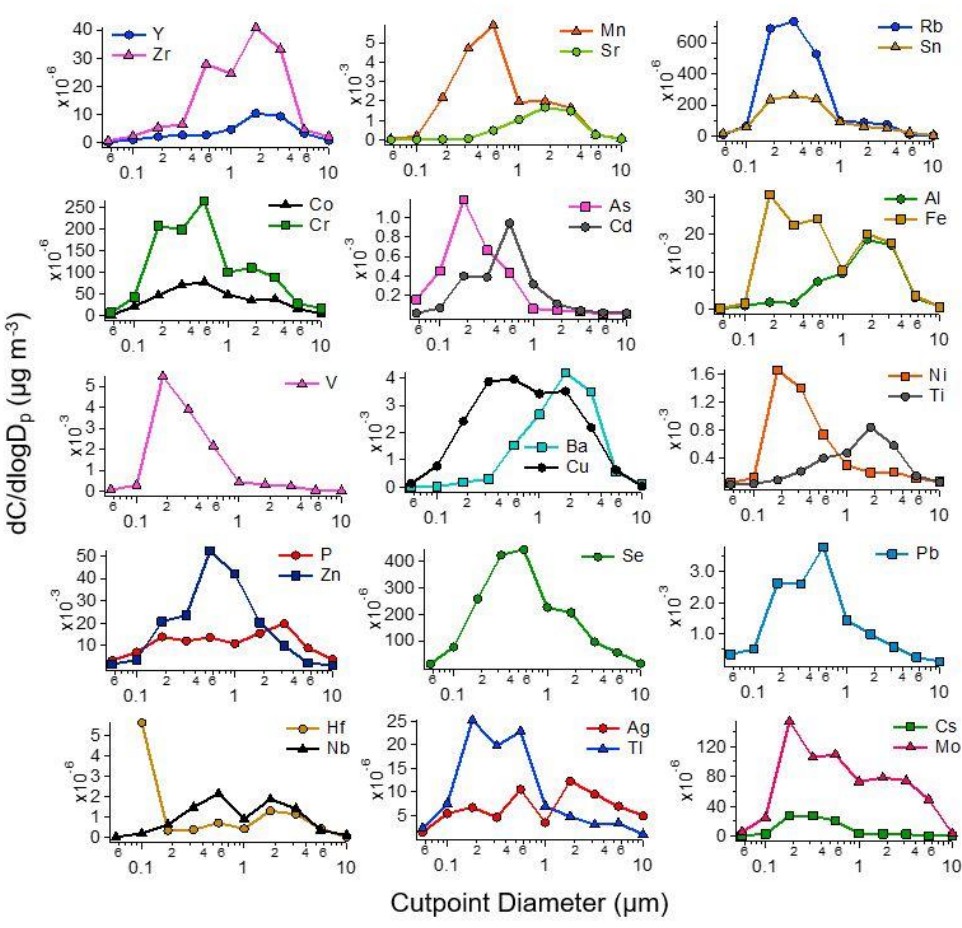

**Figure 7.** Average mass size distribution of water-soluble elements speciated via ICP-QQQ.

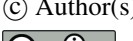



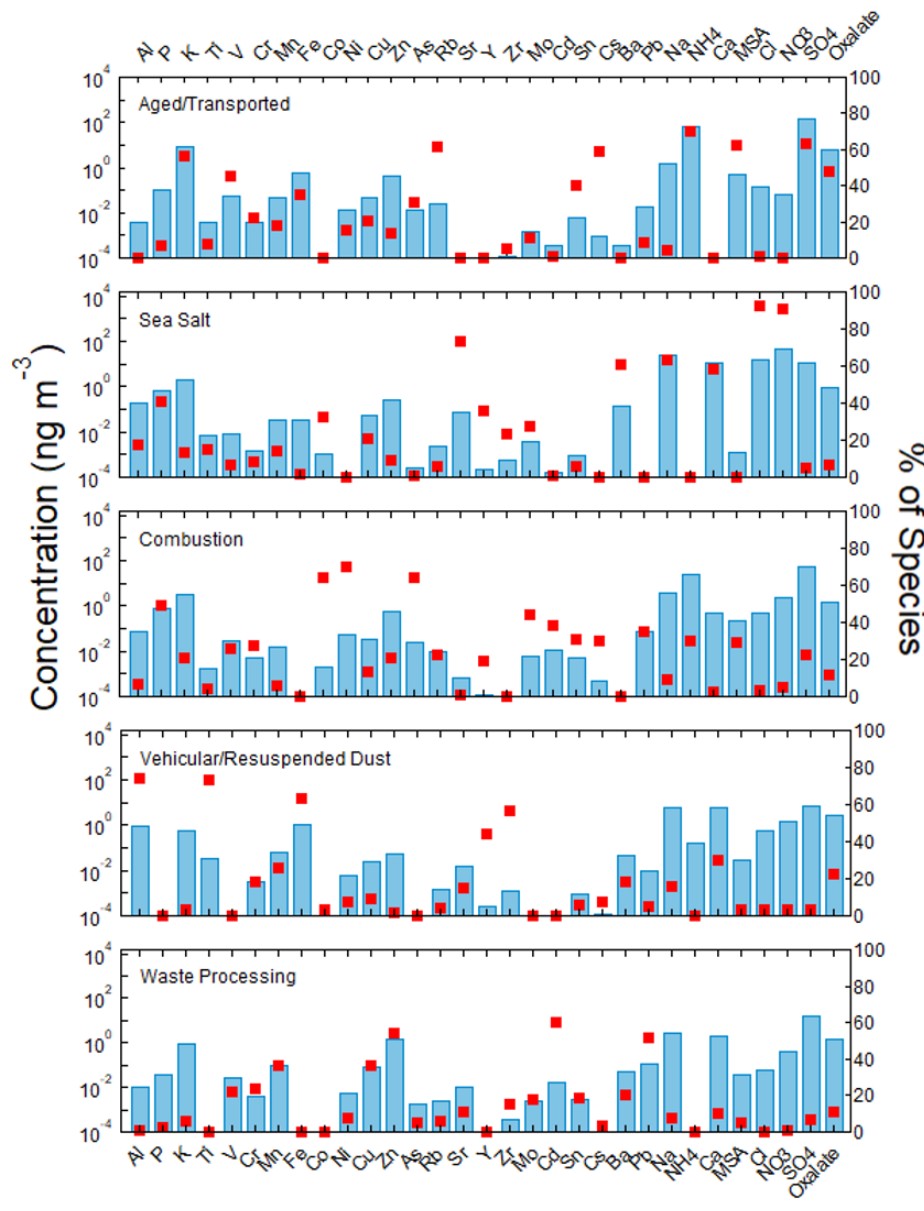

**Figure 8.** Overview of the PMF five factor solution with blue bars representing mass
concentrations and red squares signifying the percentage of mass concentration contributed to
constituents by each source factor.



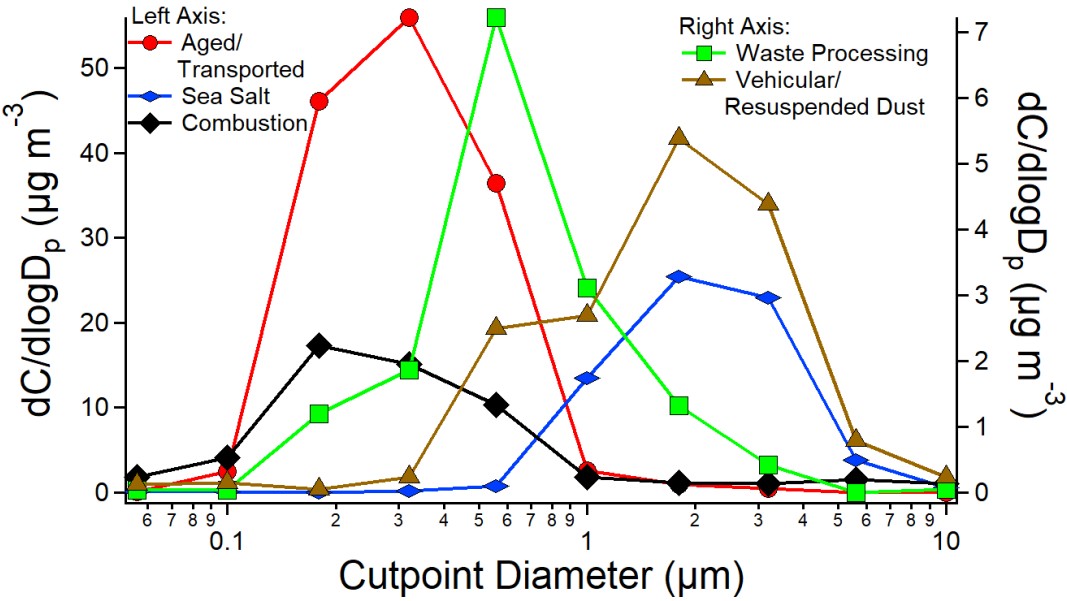

**Figure 9**. Reconstructed mass size distributions using PMF for the five major source profiles.