# Peer review of "Size-resolved Composition and Morphology of Particulate Matter During the Southwest Monsoon in Metro Manila,"

_Atmospheric Chemistry and Physics, 2019_

## Referee Comment (RC1) · Anonymous Referee #1 · 11 May 2019

The authors of this manuscript present a set of size-resolved aerosol measurements made in an urban location in Metro Manila, Philippines during the southwestern monsoon (SWM). Samples were analyzed using a variety of methods to produce average values for both gravimetric and speciated concentrations of particulate matter in twelve, roughly two-day time periods. Results are first compared with general meteorological conditions and transport pathways using a local weather station and the HYSPLIT backtrajectory model. The results were then classified into identified source types influencing the results using PMF.

The study represents a valuable addition to classification of aerosol type and sources

impacting metro Manila during the SWM, in particular through the addition of size-resolved results. The authors' methodology was generally sound and produced findings that are generally consistent with reports of similar studies in other major urban areas. However, several of their primary findings and discussion were not fully supported by the results as presented. In addition, a more thorough description of certain aspects of their experimental setup and analysis are needed to fully understand their work. As a result, I recommend publishing of the manuscript after consideration of several major comments.

Major Comments: 1. The authors present a single short duration period with one measurement of size resolved black carbon aerosol in their dataset. While this is an interesting and useful finding, they over emphasize the extent to which they can claim general representativeness of this result on the wider Manilla urban aerosol environment during the SWM. If there is a different interpretation of their results, this needs to be clarified, as it is not apparent in their results as presented. Future work with additional such measurements would be worthwhile and support the current findings, but they should take additional care to not attribute the current BC results to wider claims or conclusions.

2. The authors conclude that the most common source type identified by PMF analysis, classified as "Aged/Transported aerosol," is evidence of the influence of a major non-local source. In the conclusions they further state that "the significant presence of Aged/Transported aerosols in Metro Manila indicates that PM in the region has the ability to travel long distances during the SWM season, despite the typical assumption that wet scavenging effectively removes most of the particles." Non-local sources dominating the Manila aerosol for considerable periods of time (the authors state this factor constituted 48% of PMF classified sources by mass) would indeed constitute a considerable change in the understanding of the sources of particulate matter in Manila, but this conclusion is not supported by the analysis presented here.

Given the results as presented, it would seem this identified PMF type could just as

easily be attributed to aged aerosol that includes mixing from various local and regional sources. At the very least the authors need to consider other possible interpretations of this PMF source. Is it possible this source is merely the result of mixing from local and regional sources in circumstances where the aerosol has not been impacted by recent precipitation, and therefore has aged more than other sources?

Several open questions need to be answered before this PMF type can be attributed to a non-local source.

- What constitutes a "transported" or non-local aerosol? Smoke transported from biomass burning regions in Borneo or Sumatra that have been transported thousands of kilometers would constitute a considerably different source from an anthropogenic source hundreds of km away in other parts of the Philippines, or from regional sources in cities neighboring Metro Manila.

- Address if markers for various sources constitute actual evidence of such sources dominating the PMF type and observed aerosol, or if they merely constitute mixing of various sources with local aerosol. Do the sources associated with ocean emissions (the authors note MSA and DMA), biomass burning (K is noted), or oxalate production in fact dominate this PMF type? That would not appear to be the case in the mass loadings from Figure 8. Further, $NH_4+$ and $SO4\_2-$ are observed in other urban areas, why should they constitute evidence of non-local transport here?

- Link the source with precipitation, transport pathways, and aging times. Could precipitation scrubbing or the lack thereof account for the differences between this source and others? The authors note that the source may require aging for production of various measured species, but do not account for what amount of time might be required or if that could be accounted for by local or regional sources, complex meteorology and transport pathways in the region, or differences in precipitation scrubbing of the airmass.

- Account for the high overall mass loading being transported from a non-local source.

A consistent non-local accumulation mode aerosol plume with sufficient mass loading to account for the measured mass concentrations should be noted in other measurements (e.g. AOD).

- Validate that the aerosol is not merely aged and associated with complex mixing of local and regional sources already expected to be major sources of aerosol in the region (as correctly noted by the authors).

Classifying this aerosol type as aged aerosol would seem justified by the analysis in section 3.3.1, but it does not necessarily follow that it is therefore a non-local or non-regional source. As this is implied to be a major finding of this work, the authors may wish to revisit this analysis and potentially reframe this source as an aged aerosol with some evidence of mixing with non-local sources.

Minor comments:

1. Experimental setup and site information. Additional information is needed regarding the setup of the measurement system and weather station. In particular, the location and description of the sample inlet and any initial processing of the sample that occurred. The height of the weather station above ground level in addition to the sea level reference, and its location is needed as well (e.g. Is the third floor near the top of the building? Was the station located on the roof? Were any potential sources of bias considered such as wind steering around buildings considered?) From Figure 1 it appears the sampling location is some distance from the nearest major roadway. Has previous work using this sampling location indicated it is generally representative of the local aerosol?

2. Usage of HYSPLIT. Were there any other HYSPLIT receptor heights considered. Vertical wind shear can be considerable in the maritime continent, and can alter transport pathways considerably. Table 1 indicates primarily easterly winds. Was there any comparison between backtrajectories and local wind measurements? Were expected transport pathways or HYSPLIT trajectories related in any way to sample results? Local

measurements of wind direction, which are more indicative of local transport pathways, were correlated to mass concentrations, but not backtrajectories that are more indicative of long range transport pathways. Does this relate in any way to expected source type or location?

3. Description of data collection and usage in various analyses. The experimental methods section 2 would benefit from more clarity regarding how many data points were utilized in the various analyses, and which measurements they came from. It appears that there were 12 sampling periods during which data was collected for PMF analysis, which each consisted of 11 size bins. How was the data prepared for use in the PMF analysis?

Specific comments and technical corrections:

26. Need to specify that particle size is defined on a diameter basis. Also would help to briefly describe the complete range of sizes measurements were made across before describing results in specific size ranges.

28. The authors need to clarify that the Greenfield gap term is in reference to the reduced efficiency of precipitation scavenging of accumulation mode aerosol particles as compared to particles of other sizes. The wording of this sentence seems to imply that either accumulation mode particles exist because of the lack of precipitation scavenging, or that the Greenfield gap describes accumulation mode particles. At the very least "(the so-called Greenfield gap)" should be moved to the end of the sentence.

105/113: Manila Observatory is defined after the first use of MO as an acronym.

150: Additional detail about the typical duration and strategy for collection of sample sets would be helpful in this paragraph.

156: State the specific height of the HYSPLIT backtrajectory receptor.

Section 2.2: Again, information about the inlet and sample collection strategy is needed. Is there a general methodology that was followed for collection of the aerosol

that has been reported elsewhere? If so, this should be cited. If not, more information is needed (e.g. was rotation utilized, was sample RH monitored or expected to affect samples, were there any other expected sources of potential bias in the measurements)? More information or methodology references are also needed to fully describe methodologies for the gravimetric and optical absorption analyses.

Sections 2.3 and 2.4: Methodologies described in these section are somewhat lacking in references. The authors may want to consider including additional references with more description of both the methods and any caveats to their use for interested readers.

Section 2.5: Which measurements specifically were included in the PMF and correlation analyses, and how was the size distribution of various species included in these analyses? Also specify how many data points were used for these analyses.

334: May be useful to remind readers that jeepneys are local vehicles in common usage in Manila.

349: Need to consider the relatively few BC measurements in this study and moderate speculation about the representativeness of this single measurement unless otherwise justified.

354: The authors seem to be claiming that most BC was measured in the accumulation mode primarily as a result of the lack of precipitation scrubbing in the Greenfield gap ("95% of the BC mass is concentrated in the Greenfield gap, and thus the removal of BC due to precipitation is inefficient"). While the claim that BC is not being efficiently removed by precipitation makes conceptual sense, the emphasis on this arising due to precipitation processes is not supported by the arguments presented here. Precipitation scavenging is not the only process that affects the size distribution of particles. Coagulation, growth, and aging tend to move particles from nucleation and Aitken modes into the accumulation mode, while smaller particles typically already have minimal contributions to mass distributions. Further, there were not enough measurements of BC

size distributions (only one it would seem, sample MO13) to compare relative differ­ences between distributions in periods of more and less precipitation. In this sentence specifically, and section 3.2.1 more generally, the authors make a number of claims about the size distribution of BC that appears to be based on only one sample. While the authors may comment on the nature of the BC mass distribution data point, they should refrain from undue speculation on its cause in this case, unless it is merely to mention potential relevant processes.

445: The sentence beginning on this line is not completely clear, and the authors may wish to reword to better clarify their intent. Additional explanation of why they consider the PMF solution valid may be helpful.

447: I assume the authors meant to refer to the coefficient of determination (i.e. $r^2$) rather than the coefficient of variation (i.e. sigma/mu) here.

467: I do not understand how the authors came to this conclusion based on one mea­sured BC data point. What is the interrelationship with water soluble ions that justifies concluding they necessarily vary in concert with each other? How do you correlate BC to 15 other species? Is this just based on correlation of the size distribution? Many species exhibit a similar distribution in the accumulation mode, so that alone would not seem to justify this statement.

508: In this paragraph the authors state they expect PM in Metro Manila to be domi­nated by local sources of aerosol, and that the "Aged/Transported" pollution PMF type is in fact the largest source. Is it not reasonable to associate this source with precisely what was expected for a typical background aerosol in Manila that features complex mixing between local, regional, and distant sources that have experienced some aging before being either advected to other areas or impacted by precipitation? If mixing with distant sources is in fact detectable, that does not necessarily imply that local sources are not still the dominant source.

560: How are BC correlations being conducted? Following on the earlier comment

regarding valid data points for various analyses, how many data points and of what type are used in this correlation? Is this intended to highlight BC found in the same size bins as As and Ni?

---

## Referee Comment (RC2) · Anonymous Referee #2 · 14 Jun 2019

This manuscript describes size-resolved aerosol particle composition information from the urban Manila center of the Philippines during a relatively time-limited observing campaign. The information presented represents a very useful summary of the observations and links to local and regional source production. ?The broad Southeast Asian archipelago is subject to significant air quality hazards and regional aerosol transport, making the region a hotbed for chemical and particulate aerosol study. The topic and manuscript are thus worthy of consideration by ACP. I found the paper to be relatively strong technically and the figures very clear and legible (my technical notes are attached).

[Figure]

My recommendation is that the paper be published after minor revisions.

My primary points of concern are:

1) The paper really lacks a hypothesis. As such, it reads more as a technical report, which is ultimately fine. I suspect that the impact of this paper will be found as a strong reference set of measurements to characterize a major urban center on the eastern side of the South China Sea. But, authors would be wise to reconsider motivation and establish some binding question that makes these measurements wholly unique. To that end, and as I'll point out again below, regional transport is something that the Taiwanese groups have been looking at for decades now. Perhaps this isn't technically SE Asia. But, there is a large body of work (start with N. C. Lin) showing transport from the mainland over the ocean, chemical morphology, size information, and vertical/radiative properties.

2) I found the discussion in P18/19 to be really clumsy. This simple premise that precipitation is enough to scavenge significant portions of the regional aerosol mass is very coarse. Sure, aerosol transport correlates most strongly with dry air mass movement. But, in SE Asia, particularly seasonally, the pall is immense and omnipresent. This discussion needs to be reconsidered complete. And, again, some consideration of Taiwanese experiments looking at transport from the mainland is surely relevant context to what is being seen in the Philippines.

3) Redefine your acronyms in the Conclusions, in the event that your reader only reads those summary points and nothing else.

I found the paper to be very well written, otherwise. Good luck.

Please also note the supplement to this comment:
https://www.atmos-chem-phys-discuss.net/acp-2019-270/acp-2019-270-RC2-supplement.pdf

**Supplement:**

Atmos. Chem. Phys. Discuss., https://doi.org/10.5194/acp-2019-270 Manuscript under review for journal Atmos. Chem. Phys. Discussion started: 12 April 2019

(c) Author(s) 2019. CC BY 4.0 License.

**Size-resolved Composition and Morphology of Particulate 1**

**Matter During the Southwest Monsoon in Metro Manila, 2**

**Philippines** 3

4

Melliza Templonuevo Cruz1,2, Paola Angela Bañaga1,3, Grace Betito3, Rachel A. Braun4, Connor 5

Stahl4, Mojtaba Azadi Aghdam4, Maria Obiminda Cambaliza1,3, Hossein Dadashazar4, Miguel 6

Ricardo Hilario3, Genevieve Rose Lorenzo1, Lin Ma4, Alexander B. MacDonald4, Preciosa 7

Corazon Pabroa5, John Robin Yee5, James Bernard Simpas1,3, Armin Sorooshian4,6 8

9

10 1Manila Observatory, Quezon City 1101, Philippines

[revised manuscript text omitted]

Atmospheric Chemistry and Physics

363 species exhibited a mass concentration mode between  $0.32-0.56 \,\mu$ m, including common inorganic 364 species (SO42-, NH4+), MSA, amines (DMA, TMA+DEA), and a suite of organic acids, such as 365 oxalate, phthalate, succinate, and adipate, produced via precursor volatile organic compounds 366 (VOCs). Two organic acids with peaks in other size ranges included maleate (0.56–1 µm) and 367 pyruvate (0.1–0.18 µm). Sources of the inorganics are well documented with SO42- and NH4+ 368 produced by precursor vapors SO2 and NH3, respectively, with ocean-emitted dimethylsulfide 369 (DMS) as an additional precursor to SO42- and the primary precursor to MSA.

Precursors leading to secondarily produced alkyl amines such as DMA, TMA, and DEA 370 likely originated from a combination of industrial activity, marine emissions, biomass burning, 371 vehicular activity, sewage treatment, waste incineration, and the food industry (e.g., Facchini et 372 al., 2008; Sorooshian et al., 2009; Ge et al., 2011; VandenBoer et al., 2011); another key source of 373 374 these species, animal husbandry (Mosier et al., 1973; Schade and Crutzen, 1995; Sorooshian et al., 2008), was ruled out owing to a scarcity of such activity in the study region. Secondarily produced 375 amine salts likely were formed with SO42- as the chief anion owing to its much higher 376 concentrations relative to NO3 or organic acids/Dimethylamine was the most abundant amine 377 378 similar to other marine (Muller et al., 2009) and urban regions (Youn et al., 2015); the average concentration of DMA integrated over all MOUDI stages for all sample sets was 62.2 ng m-3 in 379 contrast to 29.8 ng m-3 for TMA+DEA. For reference, the other key cation (NH4+) participating in 380 salt formation with acids such as H2SO4 and HNO3 was expectedly much more abundant (1.64 µg 381 m-3). With regard to the competitive uptake of DMA versus NH3 in particles, the molar ratio of 382 DMA:NH4+ exhibited a unimodal profile between 0.1–1.8  $\mu$ m with a peak of 0.022 between 0.32– 383 0.56  $\mu$ 
[revised manuscript text omitted]

---

## Author Comment (AC1) · 17 Jun 2019

Response: We thank the two reviewers for thoughtful suggestions and constructive criticism that have helped us improve our manuscript. Below we provide responses to reviewer concerns and suggestions.

acp-2019-270 Size-resolved Composition and Morphology of Particulate Matter During the Southwest Monsoon in Metro Manila, Philippines Melliza Templonuevo Cruz, Paola Angela Bañaga, Grace Betito, Rachel A. Braun, Connor Stahl, Mojtaba Azadi Aghdam, Maria Obiminda Cambaliza, Hossein Dadashazar, Miguel Ricardo Hilario, Genevieve Rose Lorenzo, Lin Ma, Alexander B. MacDonald, Preciosa Corazon Pabroa,

[Figure]

John Robin Yee, James Bernard Simpas, Armin Sorooshian

Reviewer #1:

The authors of this manuscript present a set of size-resolved aerosol measurements made in an urban location in Metro Manila, Philippines during the southwestern monsoon (SWM). Samples were analyzed using a variety of methods to produce average values for both gravimetric and speciated concentrations of particulate matter in twelve, roughly two-day time periods. Results are first compared with general meteorological conditions and transport pathways using a local weather station and the HYSPLIT backtrajectory model. The results were then classified into identified source types influencing the results using PMF.

The study represents a valuable addition to classification of aerosol type and sources impacting Metro Manila during the SWM, in particular through the addition of size-resolved results. The authors' methodology was generally sound and produced findings that are generally consistent with reports of similar studies in other major urban areas. However, several of their primary findings and discussion were not fully supported by the results as presented. In addition, a more thorough description of certain aspects of their experimental setup and analysis are needed to fully understand their work. As a result, I recommend publishing of the manuscript after consideration of several major comments.

Major Comments:

1. The authors present a single short duration period with one measurement of size resolved black carbon aerosol in their dataset. While this is an interesting and useful finding, they over emphasize the extent to which they can claim general representativeness of this result on the wider Manila urban aerosol environment during the SWM. If there is a different interpretation of their results, this needs to be clarified, as it is not apparent in their results as presented. Future work with additional such measurements would be worthwhile and support the current findings, but they should take additional care to not attribute the current BC results to wider claims or conclusions.

Response: To address this comment, the authors have emphasized in many parts of the paper and whenever results of the BC measurements were discussed, that the observations were based on a single MOUDI set only. Nevertheless, visual inspection of all the MOUDI sample sets collected always show black deposits in the supermicrometer range and the filters from Stage 9 (0.18-0.32 $\mu$m) were always the blackest. This gives the authors confidence that even though BC was not quantified in all the sample sets, the trends observed in Set 13 may be representative of all the samples collected during the southwest monsoon season. We also agree with the referee comment that additional BC measurements would be important and BC observations in subsequent work will be based on a number of sample sets.

2. The authors conclude that the most common source type identified by PMF analysis, classified as "Aged/Transported aerosol," is evidence of the influence of a major non-local source. In the conclusions they further state that "the significant presence of Aged/Transported aerosols in Metro Manila indicates that PM in the region has the ability to travel long distances during the SWM season, despite the typical assumption that wet scavenging effectively removes most of the particles." Non-local sources dominating the Manila aerosol for considerable periods of time (the authors state this factor constituted 48% of PMF classified sources by mass) would indeed constitute a considerable change in the understanding of the sources of particulate matter in Manila, but this conclusion is not supported by the analysis presented here. Given the results as presented, it would seem this identified PMF type could just as easily be attributed to aged aerosol that includes mixing from various local and regional sources. At the very least the authors need to consider other possible interpretations of this PMF source. Is it possible this source is merely the result of mixing from local and regional sources in circumstances where the aerosol has not been impacted by recent precipitation, and therefore has aged more than other sources?

Response: The authors revisited the "Aged/Transported" aerosol factor from PMF and concluded that indeed, regional and local sources could have influenced this factor and we have changed the name of this factor to just "Aged Aerosol."

Several open questions need to be answered before this PMF type can be attributed to a non-local source. - What constitutes a "transported" or non-local aerosol? Smoke transported from biomass burning regions in Borneo or Sumatra that have been transported thousands of kilometers would constitute a considerably different source from an anthropogenic source hundreds of km away in other parts of the Philippines, or from regional sources in cities neighboring Metro Manila. - Address if markers for various sources constitute actual evidence of such sources dominating the PMF type and observed aerosol, or if they merely constitute mixing of various sources with local aerosol. Do the sources associated with ocean emissions (the authors note MSA and DMA), biomass burning (K is noted), or oxalate production in fact dominate this PMF type? That would not appear to be the case in the mass loadings from Figure 8. Further, NH4+ and SO4_2- are observed in other urban areas, why should they constitute evidence of non-local transport here? - Link the source with precipitation, transport pathways, and aging times. Could precipitation scrubbing or the lack thereof account for the differences between this source and others? The authors note that the source may require aging for production of various measured species, but do not account for what amount of time might be required or if that could be accounted for by local or regional sources, complex meteorology and transport pathways in the region, or differences in precipitation scrubbing of the airmass. - Account for the high overall mass loading being transported from a non-local source. A consistent non-local accumulation mode aerosol plume with sufficient mass loading to account for the measured mass concentrations should be noted in other measurements (e.g. AOD). - Validate that the aerosol is not merely aged and associated with complex mixing of local and regional sources already expected to be major sources of aerosol in the region (as correctly noted by the authors).

Response: The authors initially added "Transported" to this PMF factor to account for the presence of markers for ocean emissions and biomass burning. However, as the reviewer has pointed out, the markers for these sources (MSA, DMA, and K) do not dominate this PMF factor. The dominant species for this factor are NH4+ and SO42-, which are products of secondary particle formation from precursors and could come from local and regional sources.

Classifying this aerosol type as aged aerosol would seem justified by the analysis in section 3.3.1, but it does not necessarily follow that it is therefore a non-local or non-regional source. As this is implied to be a major finding of this work, the authors may wish to revisit this analysis and potentially reframe this source as an aged aerosol with some evidence of mixing with non-local sources.

Response: We thank the reviewer for pointing this out and we have changed this factor from "Aged/Transported" to "Aged Aerosol." The discussion in Section 3.3.1 has been edited and now reads as follows:

[revised manuscript text omitted]

The section of the Conclusions, which discussed the "Aged/Transported" aerosol factor, has been revised to read as follows:

PMF analysis suggested that there were five factors influencing the water-soluble fraction of PM collected at the sampling site. These factors, their contribution to total water-soluble mass, and the main species that permit them to be linked to a physical source are as follows: Aged Aerosol (48.0%; $NH_4^+$, $SO_4^{2-}$, MSA, oxalate), Sea Salt (22.5%; $Cl^-$, $NO_3^-$, $Ca^{2+}$, $Na^+$, $Mg^{2+}$, Ba, Sr), Combustion (18.7%; Ni, As, Co, P, Mo, Cr), Vehicular/Resuspended Dust (5.6%; Al, Ti, Fe), and Waste Processing (5.1%; Zn, Cd, Pb, Mn, Cu). The dominant contribution of Aged aerosols to water-soluble mass contradicts two expectations: (i) locally-produced sources in polluted cities should drown out the signal of transported aerosols, and (ii) the signal of transported aerosols should be significantly reduced due to scavenging processes upwind of the measurement site.

Although the current study focuses exclusively on the SWM season in Metro Manila, results of this study are applicable to the study of aerosol impacts on Southeast Asia and other regions. First, the detection of Aged aerosols not only from local but also from regional sources confirms previous studies that PM in the region has the ability to travel long distances during the SWM season. Characterization of aerosols in Metro Manila is therefore important for better understanding the impacts that local emissions will have on locations downwind of Metro Manila, including other populated cities in Southeast and East Asia. Transport of pollution and decreased wet scavenging during the SWM season may become increasingly important as studies have shown a decrease in SWM rainfall and increase in the number of no-rain days during the SWM season in the western Philippines in recent decades (e.g., Cruz et al., 2013).

Minor Comments:

1. Experimental setup and site information. Additional information is needed regarding the setup of the measurement system and weather station. In particular, the location and description of the sample inlet and any initial processing of the sample that occurred. The height of the weather station above ground level in addition to the sea level reference, and its location is needed as well (e.g. Is the third floor near the top of the building? Was the station located on the roof? Were any potential sources of bias considered such as wind steering around buildings considered?) From Figure 1 it appears the sampling location is some distance from the nearest major roadway. Has previous work using this sampling location indicated it is generally representative of the local aerosol?

Response: More details about the study site, the location of the MOUDIs and their inlets, and the location of the weather station have been added to Section 2.1. Since the MOUDI inlet was located outside a window on the southern side of the MO Administration building, the authors have considered the potential bias this could pose but we do not expect it to have a significant effect on our results especially since local wind directions (as determined by the weather station located just above the inlet) mostly came from the east and southeast directions. In terms of the local representativeness of the sampling site, source apportionment results at the Manila Observatory by Cohen et al. (2009) are in agreement with that of Kim Oanh et al. (2013), which presented the source apportionment results of data from six different sites in Metro Manila. This indicates that the aerosol collected at the Manila Observatory is representative of Metro Manila aerosol.

To address the comments above, Section 2.1, has been edited as follows:

Sampling was performed at MO in Quezon City, Philippines (14.64° N, 121.08° E). Two MOUDIs were placed inside an unoccupied room on the 3rd floor of the MO administration building (∼85 m above sea level). The inlet, located just outside the window, consists of a 2 m long stainless steel tube and a reducer that is connected directly to the MOUDI inlet. Figure 1 visually shows the sampling location and potential surrounding aerosol sources. Past work focused on PM2.5 suggested that the study location is impacted locally mostly by traffic, various forms of industrial activity, meat cooking from local eateries, and, based on the season, biomass burning (Cohen et al., 2009). This is consistent with another source apportionment study which reported that potential sources in six sites across Metro Manila include traffic, secondary particles, and biomass burning (Kim Oanh et al., 2013).

Meteorological data were collected using a Davis Vantage Pro 2 Plus weather station located on the roof (∼90 m above sea level, ∼15 m above ground level) above where the MOUDIs were located. Except for precipitation, which is reported here as accumulated rainfall, reported values for each meteorological parameter represent averages for the sampling duration of each aerosol measurement. The mean temperature during the periods of MOUDI sample collection ranged from 24.9 to 28.1° C, with accumulated rainfall ranging widely from no rain to up to 78.4 mm. To identify sources impacting PM via long-range transport to the Metro Manila region, Figure 1a summarizes the five-day back-trajectories for air masses arriving at MO on the days when samples were being collected, calculated using the NOAA Hybrid Single-Particle Lagrangian Integrated

Trajectory (HYSPLIT) model (Stein et al., 2015; Rolph, 2016). Trajectory calculations were started at 00, 06, 12, and 18 hours in MO at the height of the MOUDI inlet (∼ 12 m above ground level) using meteorological files from the NCEP/NCAR Reanalysis dataset. Trajectory cluster analysis was conducted using TrajStat (Wang et al., 2009). The back-trajectories in Figure 1a show that indeed 66% of the wind came from the southwest during the sampling periods.

2. Usage of HYSPLIT. Were there any other HYSPLIT receptor heights considered. Vertical wind shear can be considerable in the maritime continent, and can alter transport pathways considerably. Table 1 indicates primarily easterly winds. Was there any comparison between backtrajectories and local wind measurements? Were expected transport pathways or HYSPLIT trajectories related in any way to sample results? Local measurements of wind direction, which are more indicative of local transport pathways, were correlated to mass concentrations, but not backtrajectories that are more indicative of long range transport pathways. Does this relate in any way to expected source type or location?

Response: Aside from simulations at the height of the MOUDI inlet, HYSPLIT runs at 500, 1000, and 1500 m above ground level were also performed. Results of these runs consistently showed that 5-day backward trajectories were in general coming from the southwest (∼70%), contrary to local wind measurements that show winds coming mostly from the east. It was only during one occasion (Set 13/14 collected on Oct 6-8) that HYSPLIT and local measurements showed the same wind direction. This suggests that contribution from local sources are significant and this was confirmed by the source apportionment results.

3. Description of data collection and usage in various analyses. The experimental methods section 2 would benefit from more clarity regarding how many data points were utilized in the various analyses, and which measurements they came from. It appears that there were 12 sampling periods during which data was collected for PMF analysis, which each consisted of 11 size bins. How was the data prepared for use in the PMF analysis?

Response: The Methods section has been edited for clarity and the comments above have been considered particularly in Sections 2.3 and 2.5.

Section 2.3:

[revised manuscript text omitted]

Specific comments and technical corrections:

26. Need to specify that particle size is defined on a diameter basis. Also would help to briefly describe the complete range of sizes measurements were made across before describing results in specific size ranges.

Response: The first part of the abstract has been edited to:

This paper presents novel results from size-resolved particulate matter (PM) mass, composition, and morphology measurements conducted during the 2018 Southwest Monsoon (SWM) season in Metro Manila, Philippines. Micro-Orifice Uniform Deposit Impactors (MOUDIs) were used to collect PM sample sets composed of size-resolved measurements at the following aerodynamic cutpoint diameters (Dp): 18, 10, 5.6, 3.2, 1.8, 1.0, 0.56, 0.32, 0.18, 0.10, 0.056 $\mu$m. Each sample set was analyzed for mass, morphology, black carbon (BC), and composition of the water-soluble fraction.

28. The authors need to clarify that the Greenfield gap term is in reference to the reduced efficiency of precipitation scavenging of accumulation mode aerosol particles as compared to particles of other sizes. The wording of this sentence seems to imply that either accumulation mode particles exist because of the lack of precipitation scavenging, or that the Greenfield gap describes accumulation mode particles. At the very least "(the so-called Greenfield gap)" should be moved to the end of the sentence.

Response: This section of the abstract has been edited to: The bulk of the PM mass was between 0.18–1.0 $\mu$m with a dominant mode between 0.32–0.56 $\mu$m. Similarly, most of the black carbon (BC) mass was found between 0.10–1.0 $\mu$m, peaking between 0.18–0.32 $\mu$m. These peaks are located in the Greenfield Gap or the size range between 0.10–1.0 $\mu$m, where wet scavenging by rain is relatively inefficient.
105/113: Manila Observatory is defined after the first use of MO as an acronym.

Response: This section has been edited to:

Metro Manila has been drawing growing interest for PM research owing to the significant levels of black carbon (BC). A large fraction of PM in Metro Manila can be attributed to BC (e.g., ∼50% of PM2.5; Kim Oanh et al., 2006), with previously measured average values of BC at the Manila Observatory (MO) reaching ∼10 $\mu$g m-3 for PM2.5 (Simpas et al., 2014). The impacts of the high levels of BC present on human health have also received attention (Kecorius et al., 2019). Identified major sources of BC include vehicular, industrial, and cooking emissions (Bautista et al., 2014; Kecorius et al., 2017). Vehicular emissions, especially along roadways where personal cars and motorcycles, commercial trucks, and motorized public transportation, including powered tricycles and jeepneys, are plentiful. For instance, measurements of PM2.5 at the National Printing Office (NPO) located alongside the major thoroughfare Epifanio de los Santos Avenue (EDSA) were on average 72 $\mu$g m-3; this value is twice the average concentration at MO, an urban mixed site located approximately 5 km from NPO (Simpas et al., 2014).

150: Additional detail about the typical duration and strategy for collection of sample sets would be helpful in this paragraph.

Response: Details about the sampling duration and scheduling have been added in Section 2.2. This section has been edited to:

Sampling was performed at MO in Quezon City, Philippines (14.64° N, 121.08° E). Two MOUDIs were placed inside an unoccupied room on the 3rd floor of the MO administration building (∼85 m above sea level). The inlet, located just outside the window, consists of a 2 m long stainless steel tube and a reducer that is connected directly to the MOUDI inlet. Figure 1 visually shows the sampling location and potential surrounding sources. Past work focused on PM2.5 suggested that the study location is impacted locally mostly by traffic, various forms of industrial activity, meat cooking from local eateries, and, based on the season, biomass burning (Cohen et al., 2009).

156: State the specific height of the HYSPLIT backtrajectory receptor.

Response: This section has been edited to:

The mean temperature during the periods of MOUDI sample collection ranged from 24.9 to 28.1° C, with accumulated rainfall ranging widely from no rain to up to 78.4 mm. To identify sources impacting PM via long-range transport to the Metro Manila region, Figure 1a summarizes the five-day back-trajectories for air masses arriving at MO on the days when samples were being collected, calculated using the NOAA Hybrid Single-Particle Lagrangian Integrated Trajectory (HYSPLIT) model (Stein et al., 2015; Rolph, 2016). Trajectory calculations were started at 00, 06, 12, and 18 hours in MO at the height of the MOUDI inlet ($\sim$ 12 m above ground level) using meteorological files from the NCEP/NCAR Reanalysis dataset.

Section 2.2: Again, information about the inlet and sample collection strategy is needed. Is there a general methodology that was followed for collection of the aerosol that has been reported elsewhere? If so, this should be cited. If not, more information is needed (e.g. was rotation utilized, was sample RH monitored or expected to affect samples, were there any other expected sources of potential bias in the measurements)? More information or methodology references are also needed to fully describe methodologies for the gravimetric and optical absorption analyses.

Response: Additional details about the location of the sampling equipment, length of inlet, sampling schedule, and gravimetric analysis have been added to Section 2.2. This section now reads as:

PM was collected on Teflon substrates (PTFE membrane, 2 $\mu$m pore, 46.2 mm, Whatman) in Micro-Orifice Uniform Deposit Impactors (MOUDI, MSP Corporation, Marple et al., 2014). Size-resolved measurements were taken at the following aerodynamic cutpoint diameters (Dp): 18, 10, 5.6, 3.2, 1.8, 1.0, 0.56, 0.32, 0.18, 0.10, 0.056 $\mu$m.

Fourteen sample sets were collected during the SWM season (July-October 2018), with details about the operational and meteorological conditions during each sample set shown in Table 1. To determine the optimum sampling time that will collect enough sample for subsequent analyses, collection time for the first four samples ranged from 24 to 119 hours. Subsequent sampling were then fixed to 48 hours with one sample set collected every week. The sampling collection was designed to include samples from each day of the week so the collection cycled between Monday – Wednesday, Tuesday – Thursday, Wednesday – Friday, and Saturday – Monday, starting at 1400 (local time) for the weekday samples and 0500 for the weekend samples. The Teflon substrates were pretreated by washing with deionized water and air drying in a covered box. Substrates were placed and retrieved from the cascade impactor inside the laboratory in an adjacent building and transported to and from the sampling site using an impactor holder (Csavina et al., 2011). Samples are immediately placed in the freezer upon retrieval.

On two occasions, two pairs of MOUDI sets (Sets MO3/MO4 and MO13/MO14) were collected simultaneously such that one set in each pair could undergo different types of analyses. Sets 3 and 13 underwent gravimetric analysis using a Sartorius ME5-F microbalance. Substrates were conditioned for at least 24 h at a mean temperature of 20-23 °C and a mean relative humidity of 30-40% before pre- and post-weighing (U.S. Environmental Protection Agency, 2016). MOUDI set 13 was additionally examined with a Multi-wavelength Absorption Black Carbon Instrument (MABI; Australian Nuclear Science and Technology Organisation). This optically-based instrument quantifies absorption and mass concentrations at seven wavelengths between 405 and 1050 nm; however, results are reported only for 870 nm to be consistent with other studies as BC is the predominant absorber at that wavelength (e.g., Ramachandran and Rajesh, 2007; Ran et al., 2016). One additional sample set for microscopy analysis was collected for one hour on August 1 using aluminum substrates.

Sections 2.3 and 2.4: Methodologies described in these section are somewhat lacking in references. The authors may want to consider including additional references with more description of both the methods and any caveats to their use for interested readers.

Response: References of the IC and ICP-QQQ analyses have been added and this section now reads:

Note that some species were detected by both IC and ICP-QQQ (i.e., Na+, K+, Mg2+, Ca2+), and that the IC concentrations are used here for all repeated species with the exception of K+ owing to better data quality from ICP-QQQ. All IC and ICP-QQQ species concentrations for samples have been corrected by subtracting concentrations from background control samples. For more examples of the application of these methods used for substrate collection and IC/ICP analysis, the reader is referred to other recent work (Braun et al., 2017; Ma et al., 2019; Schlosser et al., 2017).

Section 2.5: Which measurements specifically were included in the PMF and correlation analyses, and how was the size distribution of various species included in these analyses? Also specify how many data points were used for these analyses.

Response: The data from all the collected samples that were analysed for water-soluble ions and elements were collectively used as one input data matrix for PMF. The size was only factored in when substitution for missing concentration data was needed. For example, if a Vanadium concentration datapoint was missing for a sample with cutpoint diameter of 18 um but V was detected in more than 50% of the samples (66 samples of different cutpoint diameters) and V S/N > 1, the geometric mean used to substitute for this missing concentration data was taken from the geometric mean of the concentrations of V for samples with cutpoint diameter of 18 um only. Correlation analyses were done between species analysed by IC (12 sets = 132 samples), ICP-QQQ (12 sets = 132 samples), and BC from Set 13.

Section 2.5 that describes the PMF methodology has been edited to:

This study reports basic descriptive statistics for chemical concentrations and correlations between different variables. Statistical significance hereafter corresponds to 95% significance based on a two-tailed Student's t-test. To complement correlative analysis for identifying sources of species, positive matrix factorization (PMF) modeling was carried out using the United States Environmental Protection Agency's (US EPA) PMF version 5. A total of 132 samples from the 12 sets analyzed for water-soluble ions and elements were used in the PMF analysis. Species concentrations were examined before being inputted to PMF. Species considered as "strong" based on high signal-to-noise ratios (S/N > 1) and those with at least 50% of the concentrations above the LOD were used in the PMF modeling (Norris et al., 2014). This resulted in a 132 (samples) × 30 (species) data matrix that was inputted to PMF.

334: May be useful to remind readers that jeepneys are local vehicles in common usage in Manila.

Response: The sentence now reads: "Kecorius et al. (2017) projected that 94% of total roadside refractory PM with number concentration modes at 20 and 80 nm was linked to jeepneys, the most popular and inexpensive mode of public transport in Metro Manila."

349: Need to consider the relatively few BC measurements in this study and moderate speculation about the representativeness of this single measurement unless otherwise justified.

Response: The Abstract and parts of Section 3 have been edited to emphasize that BC observations were based on Set 13 only and that the authors' hypothesis about the effect of inefficient scavenging on the observed size distribution is just one of the many possible explanation for the observation.

354: The authors seem to be claiming that most BC was measured in the accumulation mode primarily as a result of the lack of precipitation scrubbing in the Greenfield gap ("95% of the BC mass is concentrated in the Greenfield gap, and thus the removal of

BC due to precipitation is inefficient"). While the claim that BC is not being efficiently removed by precipitation makes conceptual sense, the emphasis on this arising due to precipitation processes is not supported by the arguments presented here. Precipitation scavenging is not the only process that affects the size distribution of particles. Coagulation, growth, and aging tend to move particles from nucleation and Aitken modes into the accumulation mode, while smaller particles typically already have minimal contributions to mass distributions. Further, there were not enough measurements of BC size distributions (only one it would seem, sample MO13) to compare relative differences between distributions in periods of more and less precipitation. In this sentence specifically, and section 3.2.1 more generally, the authors make a number of claims about the size distribution of BC that appears to be based on only one sample. While the authors may comment on the nature of the BC mass distribution data point, they should refrain from undue speculation on its cause in this case, unless it is merely to mention potential relevant processes.

Response: This section has been changed to:

A possible explanation for the large contribution of BC to PM, and the persistence of PM after rain events (Kim Oanh et al., 2006), is that the BC is not efficiently scavenged by precipitating rain drops. Small particles enter rain drops via diffusion whereas large particles enter via impaction. However, particles with a diameter in the range of 0.1–1 $\mu$m (known as the Greenfield gap) are too large to diffuse efficiently and too small to impact, and are therefore not efficiently scavenged (Seinfeld and Pandis, 2016). Absorption spectroscopy of set MO13 (Figure 2b) reveals that 95% of the BC mass is concentrated in the Greenfield gap, and thus the removal of BC due to precipitation is inefficient. The Greenfield gap contains 62 $\pm$ 11% of the total mass (calculated for MO3/MO13) and 65 $\pm$ 10% of the water-soluble mass (calculated for the other 12 MO sets). As noted earlier, BC observations discussed in this paper were based only on a single MOUDI set and the effect of inefficient scavenging in the Greenfield Gap could just be one of the many potential processes affecting BC size distribution. Subsequent work that will include BC measurements in the dry season will further investigate this hypothesis.

445: The sentence beginning on this line is not completely clear, and the authors may wish to reword to better clarify their intent. Additional explanation of why they consider the PMF solution valid may be helpful.

Response: The sentence now reads:

The PMF solution with five factors (Figure 8) was chosen because it passed the criteria of physical meaningfulness and it had a calculated ratio of Qtrue:Qexpected (1.2) that was very close to the theoretical value of 1.0.

447: I assume the authors meant to refer to the coefficient of determination (i.e. $r^2$) rather than the coefficient of variation (i.e. sigma/mu) here.

Response: The sentence has been corrected to:

There was a high coefficient of determination between measured and predicted mass concentration when summing up all species for each MOUDI stage (r2 = 0.79; sample size, n = 132), which added confidence in relying on the PMF model for source apportionment of PM.

467: I do not understand how the authors came to this conclusion based on one measured BC data point. What is the interrelationship with water soluble ions that justifies concluding they necessarily vary in concert with each other? How do you correlate BC to 15 other species? Is this just based on correlation of the size distribution? Many species exhibit a similar distribution in the accumulation mode, so that alone would not seem to justify this statement.

Response: Correlation was done on BC concentrations from 11 samples from Set 13 and each of the species consisting of 132 samples from 12 MOUDI sets. This section has been edited to:

Although BC concentrations were quantified from set MO13 only, the results showed that BC was significantly correlated (r: 0.61-0.92) with 15 species, including those mentioned above (owing to co-emission) and also a few elements that were found via PMF to be stronger contributors to the Combustion source discussed in Section 3.3.3 (Ni, Cu, As, Se, Cd, Tl, Pb).

508: In this paragraph the authors state they expect PM in Metro Manila to be dominated by local sources of aerosol, and that the "Aged/Transported" pollution PMF type is in fact the largest source. Is it not reasonable to associate this source with precisely what was expected for a typical background aerosol in Manila that features complex mixing between local, regional, and distant sources that have experienced some aging before being either advected to other areas or impacted by precipitation? If mixing with distant sources is in fact detectable, that does not necessarily imply that local sources are not still the dominant source.

Response: The authors agree with this comment and have edited this part to read as follows: Even though the PM in a heavily populated urban region, such as Metro Manila, is typically thought to be dominated by local sources of aerosols, the current PMF results show that contribution from long range transport is still discernible. This finding is contrary to the expectation that the signal of transported aerosols would be lost in the noise of locally-produced aerosols.

560: How are BC correlations being conducted? Following on the earlier comment regarding valid data points for various analyses, how many data points and of what type are used in this correlation? Is this intended to highlight BC found in the same size bins as As and Ni?

Response: The BC concentrations from Set 13 (11 samples) were correlated with the As and Ni concentrations from 12 sample sets (132 samples of different cutpoint diameters) that were analysed by IC and ICP-QQQ. The authors merely want to highlight the significant correlation of BC to the two species that significantly make up the combustion source.

Reviewer #2:

This manuscript describes size-resolved aerosol particle composition information from the urban Manila center of the Philippines during a relatively time-limited observing campaign. The information presented represents a very useful summary of the observations and links to local and regional source production. ?The broad Southeast Asian archipelago is subject to significant air quality hazards and regional aerosol transport, making the region a hotbed for chemical and particulate aerosol study. The topic and manuscript are thus worthy of consideration by ACP. I found the paper to be relatively strong technically and the figures very clear and legible (my technical notes are attached).

My recommendation is that the paper be published after minor revisions.

My primary points of concern are:

1) The paper really lacks a hypothesis. As such, it reads more as a technical report, which is ultimately fine. I suspect that the impact of this paper will be found as a strong reference set of measurements to characterize a major urban center on the eastern side of the South China Sea. But, authors would be wise to reconsider motivation and establish some binding question that makes these measurements wholly unique. To that end, and as I'll point out again below, regional transport is something that the Taiwanese groups have been looking at for decades now. Perhaps this isn't technically SE Asia. But, there is a large body of work (start with N. C. Lin) showing transport from the mainland over the ocean, chemical morphology, size information, and vertical/radiative properties.

Response: We agree with the reviewer's comment that the results of this paper will serve as a strong reference measurement of PM characterization in the region. The size-resolved PM measurements, though initially done during the southwest monsoon season only, are envisioned to shed light on why total PM2.5 levels in the study site are comparable during the dry and the wet seasons, in contrast to observations in other cities in the region. Moreover, to the authors' knowledge, the initial size-resolved BC measurements have not been done before in the study area and the results provide a valuable insight on why BC levels are very high. In addition, the PMF results also provide valuable insights on the sources of aerosol in Metro Manila. Recognizing that this work is not the first to report on long range transport of aerosol in the region, the paragraph in the Conclusions related to transport has been edited to:

Although the current study focuses exclusively on the SWM season in Metro Manila, results of this study are applicable to the study of aerosol impacts on Southeast Asia and other regions. First, the detection of Aged aerosols not only from local but also from regional sources confirms previous studies that PM in the region has the ability to travel long distances during the SWM season. Characterization of aerosols in Metro Manila is therefore important for better understanding the impacts that local emissions will have on locations downwind of Metro Manila, including other populated cities in Southeast and East Asia. Transport of pollution and decreased wet scavenging during the SWM season may become increasingly important as studies have shown a decrease in SWM rainfall and increase in the number of no-rain days during the SWM season in the western Philippines in recent decades (e.g., Cruz et al., 2013).

2) I found the discussion in P18/19 to be really clumsy. This simple premise that precipitation is enough to scavenge significant portions of the regional aerosol mass is very coarse. Sure, aerosol transport correlates most strongly with dry air mass movement. But, in SE Asia, particularly seasonally, the pall is immense and omnipresent. This discussion needs to be reconsidered complete. And, again, some consideration of Taiwanese experiments looking at transport from the mainland is surely relevant context to what is being seen in the Philippines.

Response: The authors recognize that previous Taiwanese studies have shown the transport of aerosols from the mainland and from the Indochinese peninsula. Thus,

this section has been edited to: Even though the PM in a heavily populated urban region, such as Metro Manila, is typically thought to be dominated by local sources of aerosols, the current PMF results show that contribution from long range transport is still discernible. This finding is contrary to the expectation that the signal of transported aerosols would be lost in the noise of locally-produced aerosols.

3) Redefine your acronyms in the Conclusions, in the event that your reader only reads those summary points and nothing else.

Response: The authors have redefined all acronyms on the Conclusions.

I found the paper to be very well written, otherwise. Good luck. Please also note the supplement to this comment: https://www.atmos-chem-phys-discuss.net/acp-2019-270/acp-2019-270-RC2-supplement.pdf

Response: The authors have addressed the comments in the supplement by shortening the abstract and redefining all acronyms in the Conclusions. Grammar, punctuation, and style corrections suggested by the reviewer have also been made.

Please also note the supplement to this comment:
https://www.atmos-chem-phys-discuss.net/acp-2019-270/acp-2019-270-AC1-supplement.pdf

**Supplement:**

[revised manuscript text omitted]

---

## Author Comment (AC3) · 23 Jun 2019

In addition to the authors' response that was uploaded earlier, we would like to provide a marked-up manuscript version showing the changes we made as suggested by the reviewers.

Please also note the supplement to this comment: https://www.atmos-chem-phys-discuss.net/acp-2019-270/acp-2019-270-AC3-supplement.pdf

---

## Author Response (AR2)

Response: We thank the reviewer for thoughtful suggestions and constructive criticism that have helped us improve our manuscript. Below we provide responses to the suggested revisions. Reviewer comments are in bold text, authors' response are in plain text, and modifications to the manuscript are in italics. We note that we did a final read-through as the editor suggested to check for consistency, readability and flow.

acp-2019-270
Size-resolved Composition and Morphology of Particulate Matter During the Southwest Monsoon in Metro Manila, Philippines
Melliza Templonuevo Cruz, Paola Angela Bañaga, Grace Betito, Rachel A. Braun, Connor Stahl, Mojtaba Azadi Aghdam, Maria Obiminda Cambaliza, Hossein Dadashazar, Miguel Ricardo Hilario, Genevieve Rose Lorenzo, Lin Ma, Alexander B. MacDonald, Preciosa Corazon Pabroa, John Robin Yee, James Bernard Simpas, Armin Sorooshian

**Reviewer #1:**
**1. The authors have generally addressed most of the concerns raised in the review. They have moderated their speculation on the representativeness of the single BC measurement adequately. The only remaining comment I have is regarding their discussion of the updated aged aerosol factor.**

**Section 3.3.1 and the conclusion in section 4 have been updated to re-characterize the first PMF factor as "Aged Aerosol" with less emphasis on transport from distant regions. The intent was to indicate some contributions from distant sources are still discernible among the larger influence due to local or regional sources. While not critical, I had one suggestion to improve clarity in the description in section 3.3.1. The second paragraph now states:**

**"This PMF source factor is referred to as Aged Aerosol owing to its characteristic species being linked to secondary particle formation from emissions of local and regional sources. Examples include MSA and DMA being secondarily produced from ocean-derived gaseous emissions (e.g., Sorooshian et al., 2009), and K stemming from biomass burning emissions from upwind regions such as Sumatra and Borneo (Xian et al., 2013)."**

**The first sentence implies local and regional sources are the main characteristic of the factor. The second sentence then states example include ... MSA, DMA, and K... These are examples of distant transport, not local and regional sources. The authors may wish to add an additional clarification that in addition to these primary components of the factor, some species from more distant sources were detectable, such as ... the examples.**

Response: The authors revised this section and now reads as follows:

*This PMF source factor is referred to as Aged Aerosol owing to its characteristic species being linked to secondary particle formation from emissions of regional and distant sources. The presence of $NH_4^+$ and $SO_4^{2-}$ could be attributed to precursors from various local and regional combustion sources, while MSA and DMA are secondarily produced from ocean-derived gaseous emissions (e.g., Sorooshian et al., 2009). Biomass burning emissions from distant upwind regions such as Sumatra and Borneo (Xian et al., 2013) are likely sources of K.*

**2. In addition, in the conclusion in section 4 they retain two findings:**

**"The dominant contribution of Aged aerosols to water-soluble mass contradicts two expectations: (i) locally-produced sources in polluted cities should drown out the signal of transported aerosols, and (ii) the signal of transported aerosols should be significantly reduced due to scavenging processes upwind of the measurement site."**
**The first finding follows more or less from the discussion in 3.3.1, but the second does not. The discussion did not treat the extent to which distant sources were or were not reduced during transport, just that they were detectable as part of the aged PMF factor. I would also echo the comments of reviewer 2's comment here that the authors only really correlate with rainfall at the receptor, not actual meteorology along the transport route. Perhaps the idea behind (ii) should instead be stated as a question for future work in the subsequent paragraphs rather than a finding here.**

Response:  This part has been edited to:

*The dominant contribution of Aged Aerosol to water-soluble mass contradicts the expectation that locally-produced sources in polluted cities should drown out the signal of transported aerosol from distant upwind areas.*

As suggested by the reviewer, the comment on the reduction of the signal of transported aerosols has been added to the ideas for future work and the last sentence of the paper now reads as follows:

*As the current MOUDI sampling campaign at the Manila Observatory is expected to extend for a full year, future work will focus on changes in aerosol characteristics and sources on a seasonal basis as well as scavenging processes upwind of the measurement site.*

**3. One last additional specific suggestion:**

**Section 2.1:**
**"Trajectory calculations were started at 00, 06, 12, and 18 hours in MO at the height of the MOUDI inlet (~ 12 m above ground level) using meteorological files from the NCEP/NCAR Reanalysis dataset."**

**While not critical, I typically directly state the HYSPLIT receptor height used to run the model (e.g. HYSPLIT backtrajectories used model run heights of 100 m AGL... or something similar) as actual ground level and ground level in HYSPLIT model datasets often to not agree, so giving the receptor height used is often useful for the purposes of repeatability or comparison by others.**

Response:  This part has been edited to:

*Trajectory calculations were started at 00, 06, 12, and 18 hours in MO using a model run height of 12 m above ground level and meteorological files from the NCEP/NCAR Reanalysis dataset.*

**4. Lastly, I believe Fig 9 still has a reference to "Aged/Transported Aerosol" rather than "Aged Aerosol"**

Response: Figure 9 has been correctly labeled as "Aged Aerosol."

Please find, below, the marked-up version of the manuscript.

[revised manuscript text omitted]